# Increasing Cellular Uptake and Permeation of Curcumin Using a Novel Polymer-Surfactant Formulation

**DOI:** 10.3390/biom12121739

**Published:** 2022-11-23

**Authors:** Zhenqi Liu, Alison B. Lansley, Tu Ngoc Duong, John D. Smart, Ananth S. Pannala

**Affiliations:** 1Biomaterials and Drug Delivery Research Group, School of Applied Sciences, University of Brighton, Brighton BN2 4GJ, UK; 2Vietnam Academy of Science and Technology, 18 Hoang Quoc Viet, Hanoi 100000, Vietnam

**Keywords:** curcumin, Soluplus^®^, vitamin E TPGs, cell permeability, oral bioavailability, Caco-2 cell, aqueous solubility, Longvida^®^, Nacumin^®^, cytotoxicity

## Abstract

Several therapeutically active molecules are poorly water-soluble, thereby creating a challenge for pharmaceutical scientists to develop an active solution for their oral drug delivery. This study aimed to investigate the potential for novel polymer-surfactant-based formulations (designated A and B) to improve the solubility and permeability of curcumin. A solubility study and characterization studies (FTIR, DSC and XRD) were conducted for the various formulations. The cytotoxicity of formulations and commercial comparators was tested via MTT and LDH assays, and their permeability by in vitro drug transport and cellular drug uptake was established using the Caco-2 cell model. The apparent permeability coefficients (Papp) are considered a good indicator of drug permeation. However, it can be argued that the magnitude of Papp, when used to reflect the permeability of the cells to the drug, can be influenced by the initial drug concentration (C_0_) in the donor chamber. Therefore, Papp (suspension) and Papp (solution) were calculated based on the different values of C_0_. It was clear that Papp (solution) can more accurately reflect drug permeation than Papp (suspension). Formulation A, containing Soluplus^®^ and vitamin E TPGs, significantly increased the permeation and cellular uptake of curcumin compared to other samples, which is believed to be related to the increased aqueous solubility of the drug in this formulation.

## 1. Introduction

Turmeric (*Curcuma longa*, Zingiberaceae) has a long history of use in the treatment of various medical conditions [1,2], such as coughs, inflammation, respiratory diseases, influenza, sinusitis, liver disorders, rheumatism, and abdominal pain [3,4,5]. Curcumin, a major curcuminoid that occurs naturally in turmeric, is responsible for the wide range of health benefits of turmeric. The IUPAC name of curcumin is 1,7-bis(4-hydroxy-3-methoxy phenyl)-1,6-heptadiene-3,5-dione (1E-6E). The two aryl rings in curcumin contain *ortho*-methoxy phenolic groups that are symmetrically linked to a β-diketone moiety. Curcumin exhibits a pH-dependent keto–enol tautomerism; in an acidic or neutral solution, the keto form of curcumin is predominant, while in an alkaline medium, the enol form of curcumin becomes predominant (Figure 1).

Several studies on curcumin have shown that it has anti-inflammatory [6], anticancer, antibacterial [7], anti-rheumatic [8], and antitumor effects [9]. However, the very poor solubility and extensive intestinal/hepatic metabolism of curcumin limit its oral bioavailability [1], and this is the main reason why curcumin is still not approved as a therapeutic agent. To improve curcumin’s poor oral bioavailability, different formulation strategies, such as its incorporation into nanoparticles, liposomes, micelles, micro/nano-emulsions and solid dispersions, along with its co-administration with piperine, have been investigated in animal models as well as in human supplementation studies [1].

Polymers have been used for formulation development in the pharmaceutical industry for decades; a wide range of them show remarkable performance in improving the solubility and bioavailability of poorly soluble drugs [10,11]. Polymers are often used in combination with non-ionic surfactants to promote the absorption of sparingly soluble substances. The inclusion of surfactants in the formulation could help to further improve the drug solubility of polymer-containing formulations and reduce the risk of drug precipitation [12,13,14,15].

Soluplus^®^ is a polymer that has been reported in many studies to form amorphous solid dispersions by the use of spray drying, hot-melt extrusion, electrospinning and solvent evaporation techniques [12,16,17,18,19]. It has a hydrophilic polyethylene glycol (PEG) backbone, with either one or two lipophilic sidechains consisting of vinyl acetate, randomly co-polymerized with vinyl caprolactam. The chemical structure of Soluplus^®^ is shown in Figure 2.

It was reported that Soluplus^®^ acted as an effective absorption enhancer for poorly aqueous soluble substances, such as danazol and itraconazole [11]. Soluplus^®^ was used as the carrier material for a solid dispersion formulation of curcumin and showed considerable improvements in aqueous solubility and oral bioavailability [20]. Moreover, Soluplus^®^ exhibited significant enhancement in the oral bioavailability of curcumin when in combination with non-ionic surfactants, such as Solutol^®^ HS 15 [21]. It was reported that Soluplus and curcumin do not interact when physically mixed, making it suitable to be used as an excipient for curcumin formulation [22].

Vitamin E TPGs (d-α-tocopheryl polyethylene glycol 1000 succinate) is a nonionic surfactant with an average molecular weight of 1513 g/mol. The hydrophilic/lipophilic balance value is 13.2, which means that it is more hydrophilic. It is also an FDA-approved safe pharmaceutical adjuvant. From toxicology studies in animals (rats and rabbits), the European Food Safety Authority reported an overall no-observed-adverse-effect level (NOAEL) of 1 g/kg of body weight per day [23]. Several previous studies have shown that vitamin E TPGs exhibited enhanced dissolution and oral bioavailability of drugs when used as a surfactant in polymer-surfactant solid dispersions of insoluble drugs [12,24,25].

Caco-2 cells are a human colon epithelial cancer cell line with characteristics similar to those of intestinal epithelial cells, such as the formation of polarized monolayers with well-defined brush borders and tight junctions. They are often used as a model for in vitro bidirectional transport experiments (from the apical side to the basolateral side and the basolateral side to the apical side) to predict the intestinal absorption and intestinal secretion (i.e., basal uptake and apical efflux) of drugs and compounds. The apical and basolateral chambers represent the luminal and blood/mesenteric lymph sides of the gastrointestinal tract, respectively [26,27]. It was reported by Yu et al. that the cellular uptake of curcumin into Caco-2 cells was achieved by passive diffusion [28].

In this study, two polymer-surfactant-based formulations were developed (Formulations A and B) using different techniques. Formulation A was prepared using Soluplus^®^ and Vitamin E TPGs by solvent evaporation, combined with freeze-drying, while Formulation B (a powdered formulation) was prepared from PEG400 and Poloxamer 407, using a melting and mixing method. A saturated solubility study was conducted to investigate the effect of Formulation A and Formulation B on the solubility of curcumin. Characterization studies, including FTIR (Fourier transform infrared spectroscopy), DSC (differential scanning calorimetry) and XRD (X-ray powder diffraction) were carried out to evaluate the physical and chemical properties of Formulations A and B. The effect of the formulations on the absorption of curcumin was investigated by in vitro permeability and cellular uptake tests, using a Caco-2 cell monolayer model. Pure commercial curcumin and two commercially available over-the-counter curcumin supplement products, Longvida^®^ and Nacumin^®^, were selected for comparison against Formulations A and B. Longvida^®^ is a solid lipid curcumin particle (SLCP)-based formulation developed by Verdure Sciences (Noblesville, Indiana, USA), in collaboration with scientists from the University of California (Los Angeles, California, USA) [29]. It was marketed as a product with 285× increased oral bioavailability of curcumin and is sold as capsules and tablets [30]. The curcumin content in Longvida^®^ is reported to be 20–30% *w*/*w* [31]. The health benefits of taking Longvida^®^ have been well-documented by published research studies. McFarlin et al. found that muscle inflammation in patients with exercise-induced muscle damage can be relieved by taking Longvida^®^ continuously for 3 days at a dose of 400 mg/day [32]. A double-blind, placebo-controlled clinical trial conducted by Cox et al. showed that middle-aged and older adults (aged 50–80 years) exhibited improved mood and better working memory from taking Longvida^®^ at a dose of 400 mg/day for 12 weeks [33]. In another clinical trial conducted by Santos-Parker et al., it was found that the 12-week oral administration of Longvida^®^ at a dose of 2000 mg/day can improve resistance artery endothelial function in healthy adults, who are aged between 45 and 74 years, by increasing the nitric oxide bioavailability and reducing vascular oxidative stress [34]. Furthermore, Gupte et al. found that by taking 400 mg of Longvida^®^ twice daily for 90 days, it demonstrated significant pain relief capacity in middle-aged and elderly patients (aged 40 to 65 years) who were suffering from osteoarthritis of the knee [29]. Nacumin^®^ is a new nano-curcumin formulation developed by TechBiFarm, Vietnam, in collaboration with scientists from Vietnam and the UK. It is available on the market as capsules, tablets, and water-soluble powders [35]. Currently, there are no published studies about the efficacy of Nacumin^®^. The manufacturer claims that it is a curcumin supplement product “containing super-soluble nano-size natural curcumin” [35].

## 2. Materials and Methods

### 2.1. Materials

Commercial curcumin powder was obtained from TechBiFarm (Hanoi, Vietnam). Soluplus^®^ was provided by BASF SE (Ludwigshafen, Germany). Polyethylene glycol (PEG) 400, Poloxamer 407, Vitamin E TPGs, and microcrystalline cellulose (MCC) were obtained from Sigma-Aldrich Ltd. (Gillingham, Dorset, UK). Magnesium stearate was purchased from Merck (Gillingham, Dorset, UK) and Aerosil® from Evonik (Essen, Germany). Longvida^®^ Optimized Curcumin 500 mg capsules were purchased from Igennus Healthcare Nutrition (Cambridge, UK). Nacumin^®^ capsules with an average weight of 420 mg of powder mixture per capsule were a gift from TechBiFarm (Hanoi, Vietnam).

The human colon adenocarcinoma cell line, known as Caco-2 cells, was procured from the European Collection of Authenticated Cell Cultures (Salisbury, UK). Gibco^®^ Dulbecco’s modified Eagle’s medium (DMEM), which is high in glucose and L-glutamine but without sodium pyruvate, fetal bovine serum (FBS), Gibco^®^ MEM non-essential amino acids (NEAA) solution, Gibco^®^ penicillin-streptomycin solution (10,000 U/mL) and dimethyl sulfoxide (DMSO) were purchased from Fisher Scientific Ltd. (Loughborough, UK).

Gibco^®^ Trypsin-EDTA (0.25%) solution was obtained from Merck Life Sciences UK Limited (Gillingham, Dorset, UK). Hank’s balanced salt solution without phenol red (HBSS), HEPES(4-(2-hydroxyethyl)-1-piperazineethanesulfonic acid), L-lactate lithium salt, ß-nicotinamide adenine dinucleotide (ß-NAD), 1-methoxyphenazine methosulfate (MPMS), Thiazolyl Blue tetrazolium bromide (MTT), Tris-HCI buffer (Trizma hydrochloride buffer), 1 M, pH 8.0 and Triton X-100 were purchased from Sigma-Aldrich Ltd. (Gillingham, Dorset, UK). HPLC-grade acetone, ethanol, acetonitrile and orthophosphoric acid solution were acquired from Fisher Scientific Ltd. (Loughborough, UK).

For the analysis, 12-well cell culture plates with lids and 12-well transparent membrane inserts (0.4 μm pore size) were purchased from Sarstedt (Nümbrecht, Germany). The 96-well cell culture plates were purchased from Fisher Scientific Ltd. (Loughborough, UK). The T-75 cell culture flasks were obtained from Costar (Corning Incorporated, Corning, NY, USA).

### 2.2. Methods

#### 2.2.1. Preparation of Formulations A and B

Formulation A was prepared in a two-step procedure consisting of solvent evaporation and freeze-drying. First, 0.5 g commercial curcumin powder, 5 g Soluplus^®^ and 5 g Vitamin E TPGs were added to 100 mL acetone in a round-bottomed flask. The flask was then sonicated in an ultrasonic water bath (Fisherbrand™ S-Series unheated ultrasonic cleaning bath, Fisher Scientific, Loughborough, UK) for the complete dissolution of the drug and the excipients. The flask was then attached to a rotary solvent evaporator (IKA^®^ RV-10 control with water bath and Vacuubrand diaphragm pump, Fisher Scientific, Loughborough, UK) and acetone was removed under vacuum (40 °C, 250 mbar, 200 rpm). The thin waxy film that formed subsequently in the flask was sprayed with deionized water and the flask was sonicated for 10 min to let the film detach from the inner wall of the flask. The separated thin film was transferred into a 200 mL beaker, frozen at −80 °C overnight and lyophilized for 24 h in a freeze dryer (Alpha 2-4 freeze dryer, Martin Christ, Osterode am Harz, Germany). After lyophilization, a block of golden-colored mass was obtained in the beaker. The mass was then physically crushed into granules by using mortar and pestle and sieved through a 1.4-millimeter sieve.

Formulation B was prepared in a two-step procedure involving melting and mixing. First, 2 g of commercial curcumin powder, 1 g PEG400 (water-miscible solvent) and 1.8 g Poloxamer 407 (surfactant) were weighed and transferred to a beaker. The beaker was placed on top of a stainless-steel steam water bath and slowly melted using the heat produced by the steam. The mixture was continuously stirred until it resulted in the formation of a thick golden-colored mixture. The curcumin-PEG400-Poloxamer 407 mixture was then cooled to room temperature, followed by the addition of 11.2 g of microcrystalline cellulose (MCC), included to give the formulation a more solid texture, and mixed vigorously with a spatula. To this mixture, 0.2 g of magnesium stearate and 0.2 g of Aerosil® (colloidal silicon dioxide) were added to promote granule flow and aid compression. The mixture was transferred into a mortar and mixed thoroughly using a pestle, which resulted in the formation of granules. The granules were finally sieved through a 1.4 mm sieve.

#### 2.2.2. Solubility Study

Excess amounts of Formulation A, Formulation B and commercial curcumin powder (curcumin control) were added to glass vials containing 10 mL of pH 6.8 buffer and then left on an orbital shaker at 37 °C for 24 h. First, pH 6.8 buffer (50 mM) was prepared by adding potassium phosphate monobasic (6.805 g) and sodium hydroxide (0.896 g) in 1000 mL of deionized water and adjusting the pH to 6.8 ± 0.1 with 1 N NaOH. After 24 h, 4 mL of the suspension sample was collected and filtered using a 0.45 μm syringe filter. The filtered sample was analyzed with a UV spectrophotometer (Lambda 265 UV/Vis Spectrophotometer, Perkin Elmer, Waltham, MA, USA) at 420 nm to determine the curcumin concentration. The concentration of curcumin that was detected was considered the saturated solubility of the drug from each formulation. The solubility test results that were presented are the average of three individual experiments and can be expressed as the mean ± standard deviation (*n* = 3).

#### 2.2.3. Fourier Transform Infrared (FTIR) Spectroscopy

In this study, FTIR spectroscopy was used to confirm the composition of Formulations A and B by identifying the functional group peaks shown in the FTIR spectra. FTIR analysis was performed with a PerkinElmer Spectrum 65 Infra-red spectrophotometer (UK) for commercial curcumin, PEG400, Soluplus^®^, Poloxamer 407, Vitamin E TPGs, MCC, magnesium stearate, Aerosil®, Formulation A, Formulation B and their physical mixtures. A background scan was conducted before each sample was loaded. For each sample, 8 scans were carried out at a resolution of 4 cm^−1^ and a frequency range from 4000 cm^−1^ to 400 cm^−1^. FTIR software (PerkinElmer Spectrum, version 10.03.07) was used to compare the spectra and calculate the correlations.

#### 2.2.4. Differential Scanning Calorimetry (DSC)

The thermal behavior of the formulations was determined by DSC Q2000. First, 4 mg of accurately weighed commercial curcumin, Soluplus^®^, Poloxamer 407, Vitamin E TPGs, MCC, magnesium stearate, Aerosil®, Formulation A, Formulation B and their physical mixtures were loaded separately into standard aluminum pans. The sample pans were sealed with standard aluminum lids. As for PEG400, 30 mg was loaded into an aluminum pan and sealed with a hermetic lid. An empty aluminum pan sealed with a standard aluminum lid was used as the reference for samples in solid form, while another empty pan sealed with a hermetic lid was used as the reference for the sample in liquid form (PEG400). Each sample was analyzed over a temperature range from 20 °C to 250 °C, with a heating rate of 15 °C/min and a nitrogen gas flow rate of 40 mL/min.

#### 2.2.5. X-ray Powder Diffraction (XRD)

XRD was used for detecting the crystallinity of Formulations A and B. Whether a drug exists in a crystalline or amorphous form could have a significant impact on its solubility. XRD analyses of commercial curcumin powder, Soluplus^®^, Poloxamer 407, Vitamin E TPGs, MCC, magnesium stearate, Aerosil®, Formulation A, Formulation B and their physical mixtures were performed over the range of 2θ (the angle between the transmitted X-ray beam and reflected X-ray beam) from 0° to 51.56° with an Oxford Diffraction Xcalibur microfocus NovaT X-ray diffractometer (Agilent Technologies, Oxford, UK), using Cu Kα radiation. The same amount of powder of each sample was loaded into a 0.7 mm diameter capillary, then compaction was applied to the sample-loaded capillary. Subsequently, the capillary was placed in the XRD instrument for analysis. The X-ray diffractometer was combined with a Titan CCD imaging system and the data was processed using CrysalisPro (Agilent Technologies, UK) software. PEG400 was not tested by XRD as it is a liquid when at room temperature.

#### 2.2.6. Determining Curcumin Content in Commercial Curcumin Powder, Formulation A, Formulation B, Longvida^®^ and Nacumin^®^

In order to determine the curcumin content in the samples, 5 mg of commercial curcumin powder, Formulation A and Formulation B were each weighed and placed into separate 50 mL conical flasks. To obtain powder samples of the Longvida^®^ and Nacumin^®^ capsules, the capsules were opened, then 5 mg of the powder of each product was collected from the inside and transferred into separate 50 mL conical flasks. Ethanol (50 mL) was added to each conical flask and shaken thoroughly to dissolve the curcuminoids, resulting in 0.1 mg/mL curcumin stock solutions. The stock solutions were then diluted, using a 50:50 mixture of 0.1% *v*/*v* orthophosphoric acid and acetonitrile solutions, to form 0.05 mg/mL standard solutions of curcumin, Formulation A, Formulation B, Longvida^®^ and Nacumin^®^.

The standard solutions were analyzed by HPLC, having been developed specifically for the current study, to determine the concentrations of curcumin, demethoxycurcumin and bisdemoethoxycurcumin contained in each sample. The results were expressed as the mean ± standard deviation (*n* = 6). Samples were analyzed using a SphereClone™ C-18 reverse-phase silica column (SphereClone™, 150 × 4.6 mm, 5 μm particle size) on an Agilent 1100 HPLC system (Agilent Technologies, Santa Clara, CA, USA) with a solvent cabinet, pump system, UV–visible detector and manual injection valve. For the analysis, 20 μL of each sample was injected manually into the HPLC system, using a manual syringe (Agilent, needle length 50 mm, volume 1 mL, Agilent Technologies, USA). The mobile phase consisted of a 55:45 mixture of 0.1% *v*/*v* orthophosphoric acid and acetonitrile solutions, with an isocratic flow rate of 0.8 mL/min and a run time of 15 min for each sample. Curcumin was detected at 430 nm using a UV-visible detector. Standard concentrations of curcumin over a concentration range of 1 to 100 μg/mL were prepared in ethanol in triplicate and analyzed by HPLC, showing a linear relationship with a correlation coefficient of R^2^ > 0.995 over the entire range (Appendix A).

#### 2.2.7. Caco-2 Cell Culture

Caco-2 cells were grown in T-75 cm^2^ cell culture flasks (Corning, USA) at 37 °C, with 5% CO_2_. The cell culture medium consisted of DMEM + 10% (*v*/*v*) FBS + 1% (*v*/*v*) NEAA + 1% (*v*/*v*) penicillin-streptomycin solution. The Caco-2 cells were passaged every 3 days when the cell density reached 80% confluence, at a 1:3 split ratio. During passages, the old cell medium was removed from the flasks and the cells were washed with phosphate-buffered saline (PBS). Then, 5 mL of Trypsin-EDTA (0.25% *w*/*v*) solution was added to each flask and incubated at 37 °C with 5% CO_2_ for 2–3 min to detach the cells from the flasks. After trypsinization, 5 mL of the cell medium was added and the suspension containing the Caco-2 cells was centrifuged for 5 min at 500 rpm. After removal of the supernatant, the cell pellets were added to new cell culture flasks with fresh cell medium and incubated at 37 °C with 5% CO_2_. All Caco-2 cells were used between passages 40 and 50 for the MTT assay, LDH (lactate dehydrogenase) assay, drug transport study and cellular uptake study.

#### 2.2.8. Cytotoxicity Tests (MTT and LDH Assays)

The cytotoxicity of each test sample on Caco-2 cells was measured using the MTT and LDH assays. Using the results obtained from the MTT and LDH assay results, the least cytotoxic drug concentration was selected to prepare the donor suspension used in the in vitro drug transport study. DMEM + 10% (*v*/*v*) FBS + 1% (*v*/*v*) NEAA + 1% (*v*/*v*) penicillin-streptomycin solution was used as the cell culture medium in both assays.

Suspensions of commercial curcumin, Formulation A, Formulation B, Longvida^®^ and Nacumin^®^ were prepared in the cell culture medium, at a curcumin-equivalent concentration range of 0.08 mM, 0.1 mM, 0.2 mM, 0.5 mM, 0.8 mM, and 1 mM. The sample suspensions were sonicated for 1 min by an ultrasonic processor (Cole-Parmer, Saint Neots, UK) to generate homogenized suspensions.

##### MTT Assay

Caco-2 cells were seeded at a density of 4 × 10^4^ cells/well in 50 μL cell culture medium in a 96-well plate and incubated at 37 °C, with 5% CO_2_. After 24 h, the cell culture medium was replaced with 100 μL of sample suspension. Wells containing Caco-2 cells were treated with 100 μL of DMSO to completely kill the cells; these were used as the positive control. Conversely, wells containing untreated Caco-2 cells with 100 μL of the cell culture medium were used as the negative control.

The cells were exposed to the test sample for 24 h in the incubator at 37 °C, with 5% CO_2_. Subsequently, the cell culture medium or sample suspension was removed and 50 μL of the 5 mg/mL MTT solution (prepared in PBS) was added to each well and incubated for 4 h to allow the MTT to be metabolized. Finally, the MTT solution was removed and 200 μL of DMSO was added per well. The plate was gently shaken on an orbital shaker for 20 min to dissolve the formazan crystals. The formazan concentration was measured using a spectrophotometer plate reader (Perkin Elmer, Waltham, MA, USA) at 540 nm. The MTT assay was repeated 3 times. In each assay, every sample was tested in six replicates. The percentage cell viability was calculated using the following Equation (1):(1)$$\%\;\mathrm{Cell}\;\mathrm{viability}=\frac{\mathrm{Mean}\;\mathrm{absorbance}\;\mathrm{from}\;\mathrm{sample}\;\mathrm{treated}\;\mathrm{cells}}{\mathrm{Mean}\;\mathrm{absorbance}\;\mathrm{from}\;\mathrm{untreated}\;\mathrm{cells}\;\left(\mathrm{control}\right)}\times 100$$

##### LDH Assay

The LDH substrate mixture solution (15 mL) was prepared by mixing the ingredients shown in Table 1.

Caco-2 cells were seeded at a density of 4 × 10^4^ cells/well in 50 μL cell culture medium in a 96-well plate and incubated at 37 °C with 5% CO_2_. After 24 h of incubation, the cell culture medium was replaced with 100 μL of sample suspension. In addition, 100 μL of 2% (*v*/*v*) Triton-X100 solution (prepared in the cell culture medium) was added to the wells with no sample suspension. This was to completely kill the cells so that they can be used as positive controls. Wells containing Caco-2 cells with cell culture medium and no sample suspension were used as negative controls.

The plate was incubated for 24 h, after which 50 μL of LDH substrate mixture solution was added to each well and incubated at 37 °C with 5% CO_2_ for 15 min. The absorbance was measured using a spectrophotometer plate reader (Perkin Elmer, Waltam, MA, USA) at 540 nm. The percentage of LDH release was calculated using Equation (2):(2)$$\%\;\mathrm{LDH}\;\mathrm{release}=\frac{\mathrm{Mean}\;\mathrm{absorbance}\;\mathrm{from}\;\mathrm{sample}\;\mathrm{treated}\;\mathrm{cells}}{\mathrm{Mean}\;\mathrm{absorbance}\;\mathrm{from}\;\mathrm{completely}\;\mathrm{killed}\;\mathrm{cells}\;\left(\mathrm{high}\;\mathrm{control}\right)}\times 100$$

#### 2.2.9. Measurement of Transepithelial Electrical Resistance (TEER)

The transepithelial electrical resistance (TEER) was measured at room temperature after 21 days of cell culture with an epithelial volt-ohmmeter equipped with STX2 “chopstick” electrodes (EVOM2™, World Precision Instruments, USA). The details of the 21-day cell culture are covered in Section 2.2.10. The TEER reading was also carried out before and after the in vitro drug transport experiment. Before measuring the resistance values of each well, the apical and the basolateral chambers were washed twice with pre-warmed HBSS at pH 6.5 and at pH 7.4, respectively. An apical pH of 6.5 and a basolateral pH of 7.4 were used to mimic the intestinal microclimate. The resistance values of the cell monolayers on the permeable membrane (R_TOTAL_) and the resistance values of the permeable membrane with no cells (R_BLANK_) were measured. The specific cell resistance values (R_TISSUE_) were calculated using Equation (3):R_TISSUE_ (Ω) = R_TOTAL_ (Ω) − R_BLANK_(Ω)(3)

The TEER values of the Caco-2 cell monolayers can be calculated using Equation (4):TEER (Ω cm^2^) = R_TISSUE_ (Ω) × A_MEMBRANE_ (cm^2^)(4)
where A_MEMBRANE_ is the area of the permeable membrane of the insert, which is 1.12 cm^2^. To ensure membrane integrity, Caco-2 cell monolayers with TEER values below 200 Ω cm^2^ were discarded.

#### 2.2.10. Determination of In Vitro Drug Transport

The release and transport of curcumin from Formulation A, Formulation B, Longvida^®^, Nacumin^®^ and pure commercial curcumin through Caco-2 cell monolayers was assessed in both apical-to-basolateral (A-B, absorptive) and basolateral-to-apical (B-A, secretory) directions at a curcumin-equivalent concentration of 0.1 mM. The procedure of the in vitro drug transport test was conducted according to the protocol reported by Hubatsch, Ragnarsson and Artursson [36].

Caco-2 cells were seeded onto 12 mm polycarbonate cell culture inserts (with an area of 1.12 cm^2^ and a pore size of 0.4 μm) at a concentration of 4 × 10^5^ cells per insert and placed in 12-well cell culture plates. DMEM + 10% (*v*/*v*) FBS + 1% (*v*/*v*) NEAA + 1% (*v*/*v*) penicillin-streptomycin solution was used as the cell culture medium for the cell-seeding process. The apical chamber was filled with 0.5 mL of cell suspension containing 8×10^5^ cells/mL of Caco-2 cells (prepared with the cell culture medium) and the basolateral chamber was filled with 1.5 mL cell culture medium solution. The cell culture plates were incubated at 37 °C with 5% CO_2_ for 16 h. Subsequently, the cell culture medium in the apical chamber was replaced to remove non-adherent cells to reduce the risk of multilayer formation. The plates were incubated again at 37 °C with 5% CO_2_ overnight. The following day, the cell culture medium in the apical and basolateral chambers was replaced and the plates were incubated for 21 days under the same condition. The cell medium at both compartments was replaced every second day. After 21 days, Caco-2 cell monolayers were observed under a light microscope (Zeiss Primovert, Bremen, Germany) to check the integrity of the Caco-2 cell monolayers.

The cell culture medium in both chambers was then removed and the cell monolayers on the inserts were washed twice with warmed HBSS to remove any traces of the cell culture medium. Subsequently, the cells were incubated with pH 6.5 HBSS on the apical side and pH 7.4 HBSS on the basolateral side for 15 min at 37 °C, with 5% CO_2_.

For the A-B drug transport experiments, donor suspensions of Formulation A, Formulation B, Longvida^®^, Nacumin^®^ and commercial curcumin powder at a curcumin equivalent concentration of 0.1 mM were prepared in pH 6.5 HBSS (buffered with 0.35 mg/mL sodium bicarbonate (NaHCO_3_) and 10 mM methanesulfonic acid). For the B-A drug transport experiments, suspensions were prepared in pH 7.4 HBSS (buffered with 25 mM HEPES and 0.35 mg/mL NaHCO_3_). The suspensions were sonicated for 1 min using an ultrasonic processor (Cole-Parmer, Saint Neots, UK) to generate homogenized suspensions.

The A-B drug transport experiments were conducted by adding 1.2 mL of pH 7.4 buffered HBSS to the basolateral chamber and 0.4 mL of suspension to the apical chamber. Aliquots (0.05 mL) were immediately withdrawn from each apical compartment (t = 0). The cells were incubated at 37 °C in an incubator, equipped with an orbital shaker with a speed of 100 rpm to minimize the effect of unstirred water layers [36]. A sample volume of 0.5 mL was then collected from the basolateral compartment at time points of 10, 30, 60, 90, 120 and 180 min, and the volume that was withdrawn was replaced by the same volume of pre-warmed pH 7.4 buffered HBSS. At the end of the A-B drug transport experiment, aliquots of 0.05 mL were withdrawn from the apical chamber (t = 180).

For the B-A drug transport test, 1.2 mL of pH 7.4 suspension was added to the basolateral chamber and 0.4 mL of pH 6.5 buffered HBSS was added to the apical chamber. Aliquots (0.05 mL) were immediately withdrawn from each basolateral compartment (t = 0). Once again, the cells were incubated at 37 °C in an incubator equipped with an orbital shaker with a speed of 100 rpm. A sample volume of 0.2 mL was then collected from the apical compartment at time points of 10, 30, 60, 90, 120 and 180 min, and the volume that was withdrawn was replaced with the same volume of pre-warmed pH 6.5 HBSS. At the end of the B-A drug transport experiment, aliquots of 0.05 mL were withdrawn from the basolateral chamber (t = 180) for determining the curcumin concentration.

The concentration of curcumin passing through the Caco-2 cell monolayers and the initial and final concentrations of curcumin in the donor chambers were measured via HPLC. The percentage of drug permeation and permeability coefficients (Papp) were calculated using Equations (5) and (6):(5)$$\%\;\mathrm{curcumin}\;\mathrm{permeation}=\frac{\mathrm{Curcumin}\;\mathrm{concentration}\;\mathrm{in}\;\mathrm{the}\;\mathrm{basolateral}\;\mathrm{chamber}}{\mathrm{Amount}\;\mathrm{of}\;\mathrm{drug}\;\mathrm{added}\;\mathrm{to}\;\mathrm{the}\;\mathrm{apical}\;\mathrm{chamber}}\times 100$$
(6)$$\mathrm{Papp}=\frac{\mathrm{dQ}}{\mathrm{dt}}\times \frac{1}{{A\;C}_{0}}$$
where $\frac{\mathrm{dQ}}{\mathrm{dt}}$ is the change in concentration in the basolateral chamber over time (μg/mL∙s); A is the surface area of the permeable membrane of the cell culture inserts (cm^2^); C_0_ is the initial drug concentration in the apical chamber (μg/mL). The in vitro drug permeation experiments for all the tested samples were repeated 3 times on different days. For each drug permeation experiment, each sample was tested in triplicate. The results presented in this study were the averages of three experiments and were expressed as the mean ± standard deviation (*n* = 9). The efflux ratio, an index to estimate the magnitude of drug efflux during the drug transport, was calculated using Equation (7):(7)$$\mathrm{Efflux}\;\mathrm{ratio}\;\left(\mathrm{ER}\right)=\frac{\mathrm{Papp}\;\left(B-A\right)}{\mathrm{Papp}\;\left(A-B\right)}$$

#### 2.2.11. Cellular Uptake Study

The cell samples used for the in vitro drug transport study were analyzed further to determine the drug uptake by the cells. After completing the in vitro drug transport experiment, the Caco-2 cell monolayer of each 12-well transparent membrane insert was washed twice with phosphate-buffered saline (pH 7.4), followed by the addition of 0.5 mL of trypsin EDTA solution. The insert was then incubated for 10 min at 37 °C to allow the Caco-2 cells to detach. This was followed by the addition of 0.5 mL of the cell medium; the suspension containing the Caco-2 cells was centrifuged for 5 min at 500 rpm to separate the cells from trypsin EDTA. Subsequently, the supernatant was removed and an aliquot (1 mL) of DMSO was added to the cell pellet in each centrifuge tube, to dissolve any curcumin that had accumulated in the Caco-2 cells. The Caco-2 cell suspension was then centrifuged at 1000 rpm for 5 min. The solvent layer containing curcumin was then separated from the cell lysate and transferred to a glass vial, after which DMSO was evaporated to dryness using nitrogen. An aliquot of 1 mL of ethanol was then added to fully dissolve the curcumin that had accumulated in the cells, diluted to 1:1, using mobile phase solution, and analyzed by HPLC to determine the cellular uptake of curcumin for each test sample. The results presented in this study were the average of three individual experiments and are expressed as mean ± standard deviation (*n* = 9).

### 2.3. Statistical Analysis

Statistical analysis of the data from the MTT assay, the LDH assay, the drug permeation concentration at each time point, C_0_(Suspension), and the cellular drug uptake were carried out by a one-way ANOVA with a post hoc Tukey test. Paired *t*-tests were used for the statistical analysis of TEER data, Papp (suspension) and Papp (solution) data. All statistical analysis was conducted using the IBM SPSS Statistics software (Version 24.0; IBM Corp, Armonk, NY, USA).

## 3. Results

### 3.1. Solubility Study

The saturation solubilities of curcumin, as recorded from Formulation A, Formulation B and commercial curcumin powder, are shown in Table 2. Formulation A and Formulation B showed approximately an 880-fold and 312-fold higher solubility of curcumin than the curcumin control.

### 3.2. Fourier Transform Infrared Spectroscopy (FTIR)

Figure 3 shows the FTIR spectra of curcumin, Soluplus^®^, Vitamin E TPGs, Formulation A and the physical mixture of curcumin, with the excipients being used for preparing Formulation A. Commercial curcumin (Figure 3a) showed its characteristic peaks at 3509 cm^−1^ (phenolic O-H stretching), 1626 cm^−1^ (aromatic C=C stretching), 1601 cm^−1^ (C=C-C, aromatic ring-stretching), 1505 cm^−1^ (C=O and C=C stretching), 1427 cm^−1^ (olefinic C-H bending), 1273 cm^−1^ (aromatic C-O stretching), and 1114 cm^−1^ (alkyl-substituted ether, C-O-C stretching) [37]. Soluplus^®^ (Figure 3b) showed its characteristic peaks at 2860 cm^−1^ (aliphatic C–H stretching), 1732 cm^−1^ (ester C=O stretching) and 1623 cm^−1^ (amide group, C(O)N) stretching) [38]. The Vitamin E TPGs spectrum (Figure 3c) shows peaks at 3467 cm^−1^ (-OH stretching), 2885 cm^−1^ (aliphatic C-H stretching), 1736 cm^−1^ (ester C=O stretching), 1465 cm^−1^ (aromatic ring, C=C-C stretching), 1240 and 1104 cm^−1^ (ether, C-O-C stretching) [39]. These observations of the characteristic peaks agree with the published literature [37,38,39]. The representative peaks of Soluplus^®^ (1730 cm^−1^, C=O stretching) and Vitamin E TPGs (2883 cm^−1^, C-H stretching) can be seen in the spectrum of Formulation A (Figure 3d) and its physical mixture (Figure 3e). An OH peak of curcumin was observed in the spectrum of the physical mixture (Figure 3e) but was absent in Formulation A (Figure 3d).

Figure 4 shows the FTIR spectra of PEG400, Poloxamer 407, microcrystalline cellulose, Aerosil®, magnesium stearate, Formulation B and a physical mixture of Formulation B. In the spectrum of PEG400 (Figure 4a), this showed characteristic peaks at 3444 cm^−1^ (O-H stretching), 2865 cm^−1^ (alkyl C-H stretching), 1456 cm^−1^, 1349 cm^−1^, 1295 cm^−1^, 1248 cm^−1^ (alkyl C-H bending) and 1094 cm^−1^ (ether C-O-C stretching) [40,41]. Poloxamer 407 (Figure 4b) exhibited peaks at 2880 cm^−1^ (C-H alkyl stretching), 1466 cm^−1^ (C-H alkyl bending), 1341 cm^−1^ (in-plane O-H bending) and 1098 cm^−1^ (ether C-O-C stretching) [42]. Microcrystalline cellulose (MCC) (Figure 4c) showed characteristic peaks at 3329 cm^−1^ (O-H stretching), 2891 cm^−1^ (C-H alkyl stretching), 1631 cm^−1^ (C=O stretching) and 1315 cm^−1^ (C-O stretching) [43]. Aerosil® (Figure 4d) exhibited a sharp peak at 1090 cm^−1^ (Si-O-Si stretching) [44]. The spectrum of magnesium stearate (Figure 4e) showed peaks at 3252 cm^−1^ (O-H stretching), 2916 cm^−1^, 2850 cm^−1^ (C–H stretching vibration), 1571 cm^−1^ and 1466 cm^−1^ (COO-stretching vibration) [45]. The values of the characteristic peaks for these excipients correspond to the results in the published literature [40,41,42,43,44,45]. The FTIR spectra of Formulation B and the physical mixture are shown in Figure 4f,g, respectively.

### 3.3. Differential Scanning Calorimetry (DSC)

The DSC thermograms of Formulation A and B were compared with commercial curcumin, the excipients, and their physical mixtures to identify any changes in the thermal properties.

Commercial curcumin powder (Figure 5a) exhibited a sharp, single endothermic peak at 176.57 °C, which corresponded to its melting point [46]. Since a sharp endotherm is expected from a highly crystalline compound, it is suggested that commercial curcumin powder is likely to have a crystalline structure [47]. The DSC thermogram of Soluplus^®^ (Figure 5b) exhibited a broad glass transition peak around 70 °C [48]. The DSC curve of Vitamin E TPGs (Figure 5c) exhibited a sharp endothermic peak at 35.48 °C, corresponding to its melting point [39]. The observed thermograms agree with the results in the published literature. The thermogram for Formulation A (Figure 5d) revealed the presence of a weaker endothermic peak at 33.50 °C, corresponding to Vitamin E TPGs. As for the physical mixture, its DSC thermogram showed a similar pattern to Formulation A; however, here, an endothermic peak was observed at 35.80 °C (Figure 5e).

Poloxamer 407 (Figure 6a) showed a sharp endothermic peak at 55.67 °C [49]. Magnesium stearate (Figure 6b) showed two endothermic peaks. The peak at 105.08 °C was due to the evaporation of moisture, while the peak at 126.59 °C indicated that the mesophase transition occurred in magnesium stearate [50]. MCC (Figure 6c) showed glass transition point peaks at 119.79 °C [51,52]. No obvious peaks were observed from PEG400 (Figure 6d) and Aerosil® (Figure 6e).

In the thermogram of Formulation B (Figure 6f), a weak endothermic peak at 45.95 °C was observed, corresponding to the melting point of Poloxamer 407. The peak shifted to a lower temperature and became broader, compared with the thermograms of Poloxamer 407. The DSC thermogram of the physical mixture is very similar to Formulation B, with only one endothermic peak observed at 44.20 °C (Figure 6g).

### 3.4. X-ray Powder Diffraction (XRD)

The XRD spectra of commercial curcumin powder, Soluplus, Vitamin E TPGs, Formulation A and its physical mixture are shown in Figure 7. Commercial curcumin (Figure 7a) exhibited several distinct diffraction peaks at 7.94°, 8.93°, 12.26°, 14.64°, 15.9°, 17.35°, 18.26°, 19.55°, 21.48°, 23.47°, 24.65°, 25.68°, 27.46° and 29.18°. In the case of Soluplus^®^ (Figure 7b), it showed a distinct intense peak at 22.69° and several less intense peaks at 8.99°, 12.28°, 15.56°, 17.43°, 18.33°, 19.29°, 25.64°, and 34.73°. Vitamin E VTPGs (Figure 7c) showed two distinct sharp peaks at 19.26° and 23.31°. Some broad peaks were observed at 26.59°, 32.9° and 36.1°. For Formulation A (Figure 7d), two sharp peaks were found at 19.2° and 23.28°. Several broad peaks can be seen at 9.09°, 14.86°, 19.1°, 26.39° and 35.98°. The physical mixture of Formulation A (Figure 7e) showed sharp peaks at 7.94°, 8.95°, 12.26°, 14.65°, 15.22°, 15.9°, 17.44°, 18.29°, 19.25°, 21.27°, 23.31°, 24.65°, 25.68°, 26.17°, 26.91°, 27.41° and 29.1°.

The XRD spectra of poloxamer 407, Aerosil®, magnesium stearate, MCC, Formulation B and its physical mixture are shown in Figure 8. Poloxamer 407 (Figure 8a) exhibited two distinct sharp peaks at 19.34° and 23.52°. Two less intense sharp peaks are shown at 26.23° and 33.16°. Aerosil® (Figure 8b) showed several intense peaks at 21.03°, 25.75°, 26.24°, 26.29°, 28.3° and 29.11°. Several less intense peaks were found at 30.18°, 31.59°, 34.58°, 36.26°, 38.12°, 40.88° and 43.01°. Various sharp peaks were observed in magnesium stearate (Figure 8c) at 9.98°, 10.28°, 18.68° and 19.96°. Some less intense peaks can be seen at 23.03°, 24.97°, 25.62°, 26.12°, 26.93°, 28.44°, 28.93°, 30.55°, 34.3° and 36.11°. MCC (Figure 8d) demonstrated a distinct sharp peak at 22.61° and a broad peak at 15.6°. Less intense sharp peaks were found at 12.32°, 18.24°, 19.06°, 25.69° and 34.65°. Formulation B (Figure 8e) showed a distinct sharp peak at 22.54°, a broad peak at 15.32°, and several less intense sharp peaks at 5.48°, 8.88°, 12.26°, 17.15°, 18.11°, 19.1°, 25.2° and 34.69°. The physical mixture of Formulation B (Figure 8f) showed a distinct sharp peak at 22.65°, a broad peak at 15.37°, and several less intense peaks at 5.47°, 8.90°, 12.21°, 17.20°, 18.09°, 19.08°, 25.64° and 34.65°.

### 3.5. Determining the Curcumin Content in Commercial Curcumin Powder, Formulation A, Formulation B, Longvida^®^ and Nacumin^®^

The percentage of curcumin contained in commercial curcumin powder, Formulation A, Formulation B, Longvida^®^ and Nacumin^®^ was calculated and is shown in Table 3. This provides information about how much of each sample was needed to prepare the drug suspension with a 0.1 mM-equivalent concentration of curcumin (see Table 4).

### 3.6. Cytotoxicity Tests (MTT and LDH Assays)

As shown in Figure 9, the MTT assay results showed that the cell viability of Caco-2 cells decreased in a curcumin concentration-dependent manner. The cell viabilities were higher than 80% when the curcumin concentration was at 0.1 mM and 0.08 mM. There was no significant difference in cell viability at curcumin concentrations of 0.08 mM and 0.1 mM (*p* > 0.05) for all test samples.

The percentage of LDH released into the cell culture medium was used as an indicator of cell death. As shown in Figure 10, the percentage of LDH released from Caco-2 cells increased in a curcumin concentration-dependent manner. The effect of the sample on the LDH release from Caco-2 cells was at a minimum when the curcumin concentration was ≤ 0.1 mM. Formulation A, Formulation B, Nacumin^®^, Longvida^®^ and curcumin suspension showed no significant difference in LDH release at 0.08 mM and 0.1 mM (*p* > 0.05).

### 3.7. In Vitro Drug Transport Experiments and the Measurement of Transepithelial Electrical Resistance (TEER)

In this study, the in vitro permeation of curcumin was studied using Caco-2 cell monolayers. After 21 days of incubation, a thin layer of cells was formed on the transparent membrane of the inserts. Observation under a light microscope (Zeiss Primostar 1 Upright Compound Microscope, Germany) revealed a well-formed cylindrical polarized monolayer of Caco-2 cells (Figure 11). The Caco-2 cells showed an average TEER value of 299 ± 52 Ω cm^2^ during the 21 days of cell seeding, indicating that the cells had formed a functional barrier. Those cells with TEER values of below 200 Ω cm^2^ were discarded due to the damage to membrane integrity.

As shown in Figure 12, the concentration of curcumin passing through the Caco-2 cell monolayers and being transported into the basolateral side from A-B was seen to increase with the exposure time for Formulation A, Formulation B, Longvida^®^ and Nacumin^®^. At the end of the transport experiment (t = 180 min), Formulation A showed the highest curcumin permeation concentration of 0.49 ± 0.07 μg/mL (equivalent to 0.00133 ± 0.00019 mM), followed by Formulation B, with 0.11 ± 0.03 μg/mL (equivalent to 0.00030 ± 0.00008 mM), Nacumin^®^, with 0.09 ± 0.01 μg/mL (equivalent to 0.00024 ± 0.00003 mM) and Longvida^®^ with 0.04 ± 0.00 μg/mL (equivalent to 0.00011 ± 0.00001 mM) of curcumin concentration. Conversely, considering curcumin transport in the B-A direction, Formulation A was the only sample that showed curcumin transport, with a cumulative total of 0.19 ± 0.04 μg/mL (equivalent to 0.00052 ± 0.00011 mM) (Figure 5). Curcumin transport was not detected from commercial curcumin in either the A-B or B-A directions.

In the statistical analysis using a one-way ANOVA, the results showed that there were significant differences among the tested samples in terms of the concentration of curcumin permeation, either from A to B (*p* ≤ 0.05) or B to A (*p* ≤ 0.05). The post hoc Tukey test results revealed that Formulation A had significantly higher A-B drug permeation concentration than all the other tested samples at every time point (*p* ≤ 0.05). Conversely, there were no significant differences between the A to B drug permeation profiles of Formulation B and Nacumin^®^ at all the time points (*p* > 0.05). Longvida^®^ showed significantly lower A-B drug permeation concentrations than Formulation B and Nacumin^®^ at almost every time point (*p* < 0.05), with no significant differences when at 60 min (*p* ≥ 0.05) (Figure 4). There was no statistically significant difference among the B to A drug permeation profiles of commercial curcumin, Formulation B, Longvida^®^ and Nacumin^®^ at every time point since curcumin permeation was not detected (*p* > 0.05). In contrast, the permeation of Formulation A was significantly higher than all the other test samples at every time point throughout the study (*p* ≤ 0.05) (Figure 13).

The TEER of the Caco-2 cell layers in the in vitro transport study was measured before and after each experiment. As shown in Table 5, at the end of the drug transport experiment, the TEER readings significantly decreased (*p* ≤ 0.05) for all the test samples.

The values of the initial curcumin concentration dissolved in the donor chamber of each tested sample (C_0_ (suspension)) are listed in Table 6. Formulation A showed the highest C_0_ (suspension), (A-B) of 17 ± 2.95 μg/mL and C_0_ (suspension), (B-A) of 15.51 ± 1.46 μg/mL, which were significantly higher than other tested samples (*p* ≤ 0.05). Formulation B exhibited the second highest of C_0_ (suspension), (A-B) of 3.72 ± 0.71 μg/mL and C_0_ (suspension), (B-A) of 1.88 ± 0.13 μg/mL, which were significantly higher than Longvida, Nacumin^®^ and commercial curcumin (*p* ≤ 0.05). No significant differences between the Longvida, Nacumin^®^ and commercial curcumin in terms of C_0_ (suspension), (A-B) (*p* > 0.05) and C_0_ (suspension), (B-A) (*p* > 0.05).

### 3.8. Determination of Permeability Coefficient (Papp)

Papp measured in Caco-2 cells is considered a good indicator of drug permeation and a predictor for the oral absorption of passively absorbed drugs in humans [53,54]. In this study Papp was calculated using two different values for C_0_ (initial drug concentration in the apical chamber):Papp (suspension) was calculated based on C_0_ which represents the initial concentration of curcumin dissolved in the suspension measured in the apical chamber. Here C_0_ is referred to as C_0_ (suspension). C_0_ (suspension) for each test sample is listed in Table 6.Papp (solution) was calculated based on C_0_ which represents the total applied dose of curcumin for each test sample. Here C_0_ is referred to as C_0_ (solution).

The value of C_0_ (suspension) for each sample differs because it is associated with the amount of curcumin solubilized in each test sample and as determined previously each test sample has different curcumin solubility. On the other hand, C_0_ (solution) was the same for every test sample since they all had the same nominal applied dose of curcumin (0.1 mM, equivalent to 36.84 μg/mL).

As shown in Table 7, the two types of Papp showed the exact opposite trend. The value of Papp (suspension) was significantly higher than Papp (solution) in both A-B (Figure 14) and B-A (Figure 15) directions (*p* ≤ 0.05). This is reasonable because the value of C_0_ used for calculating Papp (suspension) was much lower than that for calculating Papp (solution).

For Papp (suspension), (A-B), Nacumin^®^ showed the highest value of 19.58 ± 3.66 × 10^−6^ cm/s, followed by Longvida^®^ of 10.79 ± 1.42 × 10^−6^ cm/s, Formulation A of 2.52 ± 0.68 × 10^−6^ cm/s, and finally Formulation B of 2.43 ± 0.55 × 10^−6^ cm/s. Formulation A with the highest A-B drug permeation concentration showed the second lowest value of Papp (suspension), (A-B), while Longvida^®^ and Nacumin, the two formulations with the lowest A-B drug permeation concentrations, showed Papp values that were 4.29- and 7.78-fold higher than Formulation A. Formulation B had the second-highest drug permeation but showed the lowest Papp (suspension), (A-B). It was clear that the trend exhibited by the Papp (suspension) results was contradicted by the results of the drug permeation concentration. The reason for this confusing phenomenon is that Papp (suspension) reflects the rate of permeation of the drug, driven by the concentration dissolved in the donor compartment rather than the nominally applied drug dose. For instance, Nacumin^®^ and Longvida^®^ have much lower values of Papp (solution) compared to other tested samples due to their limited drug solubility in the donor chamber. On the other hand, they showed much higher values of Papp (suspension) than other tested samples, which contradicted the results of the drug permeation concentration. This was because the ratio between the amounts of drug permeation to the amount of drug dissolved in the donor compartment from Longvida^®^ and Nacumin^®^ were higher than that of any other samples, which gave them much higher values of Papp (suspension).

In contrast, Formulation A showed the highest Papp (solution), (A-B) value of 1.12 ± 0.15 × 10^−6^ cm/s, followed by Formulation B of 0.24 ± 0.07 × 10^−6^ cm/s, Nacumin^®^ of 0.21 ± 0.03 × 10^−6^ cm/s and Longvida^®^ with the lowest Papp value of 0.09 ± 0.01 × 10^−6^ cm/s. It was clear that Papp (solution) reflects the permeability of the cells to the drug from the applied dose of the drug contained in each test sample and it showed a trend consistent with the data of drug permeation concentrations, i.e., test samples with a higher drug permeation concentration showed higher Papp (solution) values. As a result, Papp (solution) was used as the indicator for the drug permeability and the predictor of in vivo drug intestinal absorption in this study.

Zero curcumin permeation from B to A was detected from all tested samples except for Formulation A. For Formulation A, the values of Papp (suspension) (B-A) were 1.00 ± 0.19 × 10^−6^ cm/s, and Papp (solution), (B-A) 0.42 ± 0.10 × 10^−6^ cm/s.

The efflux ratio, defined as the ratio of Papp in the B-A direction to the Papp in the A-B direction, was used to estimate the magnitude of drug efflux. It can also give information about what transport pathway might be involved during drug transport. The value of the efflux ratio of Formulation A was calculated to be 0.38. Other tested samples all had an efflux ratio of 0 since curcumin was not detected in the receiver chamber in the drug transport test from the B-A direction.

### 3.9. Cellular Uptake Study

Accumulation of curcumin in the Caco-2 cells was measured in the cells at the end of the in vitro drug transport experiment. The cellular uptake results showed (see Figure 16) the amount of drug absorbed by the cells but not reaching the basolateral side of the cell monolayer. It provided a prediction regarding the amount of drug that crossed the epithelial cell membrane but that has not yet entered the blood/enteric lymph of the gastrointestinal tract.

In the case of drug cellular uptake, measured after the transport experiment from A to B, Formulation A had the highest A-B curcumin accumulation of 0.46 ± 0.09 μg/mL, which was significantly higher than in any other test samples (*p* ≤ 0.05). Formulation B exhibited the second-highest A-B cellular drug accumulation with 0.10 ± 0.02 μg/mL, followed by Longvida^®^ with 0.04 ± 0.01 μg/mL, Nacumin^®^ with 0.02 ± 0.00 μg/mL and, finally, commercial curcumin with 0.02 ± 0.00 μg/mL. There were no significant differences between the A to B cellular drug accumulation of Longvida, Nacumin^®^ and commercial curcumin (*p* > 0.05).

As for drug cellular uptake measured after the transport experiment from B to A, they were all significantly lower than that measured after the drug transport experiment from A to B (*p* ≤ 0.05). Formulation A exhibited a B-A cellular drug accumulation of 0.12 ± 0.04 μg/mL, which was significantly higher than any other test samples (*p* ≤ 0.05). Formulation B, Nacumin, Longvida^®^ and commercial curcumin exhibited B-A cellular drug accumulation of 0.05 ± 0.02 μg/mL, 0.06 ± 0.02 μg/mL, 0.04 ± 0.01 μg/mL and 0.02 ± 0.01 μg/mL, respectively. There were no significant differences between their B-A cellular drug accumulation values (*p* > 0.05).

## 4. Discussion

### 4.1. Solubility Study

By definition, drug solubility is the maximum concentration of a substance that can be completely dissolved in a given solvent at a certain temperature and pressure level [55]. For orally administered drugs, they must dissolve in the gastrointestinal tract before they can be absorbed and circulated in the bloodstream [56]. Solubility is a critical factor that determines the oral bioavailability of drugs. In fact, poor drug solubility is one of the most frequent causes of low oral bioavailability [57].

Formulation A has demonstrated better results than Formulation B in terms of increasing the solubility of curcumin. It has been confirmed by the XRD results that the combination of Soluplus^®^ and Vitamin E VTPGs converted crystalline curcumin into an amorphous form. This capability was also demonstrated in another formulation, with the Soluplus^®^-Vitamin E TPGs solid dispersion of valsartan [12]. Amorphous drugs are generally better dissolved than their crystalline counterparts, due to the lack of a uniform molecular arrangement, which provides them with higher molecular mobility and greater free energy [58]. The formulation of micelles from surfactant could be another reason for the increase in curcumin solubility. Vitamin E TPGs have a lower CMC than Soluplus^®^, which means that they can form micelles more readily in aqueous media; this ability can help to solubilize poorly soluble drugs in the micellar core [59].

### 4.2. Fourier Transform Infrared Spectroscopy (FTIR)

The composition of Formulation A and Formulation B was detected by identifying and comparing the different functional peaks in the FTIR spectra. In the spectra of Formulation A (Figure 3d) and Formulation B (Figure 4f), the characteristic peak of curcumin (O-H group, 3200–3500 cm^−1^) cannot be seen. The absence of the characteristic peak of curcumin might suggest that curcumin molecules are entrapped by the excipients rather than adsorbed on the surface of the excipients; thus, the characteristic signals of curcumin were hidden [60]. An alternative explanation is that the curcumin content was too low a level to be detected by the instrument. On the other hand, the OH peak of curcumin was observed in the spectrum of the physical mixture of Formulation A (Figure 3e) but was absent in the spectrum of the physical mixture of Formulation B (Figure 4g). A possible explanation for the absence of the OH peak of curcumin in the physical mixture of Formulation B is that the curcumin was dissolved in PEG400 during the preparation of the physical mixture so that its signal could not be detected. PEG400 was in liquid form at room temperature; it was found that curcumin is soluble in PEG400, with a solubility of 9.92 ± 0.24 mg/mL [61]. Another possible explanation is that there may have been an inhomogeneous mixing of the drug and excipients during the preparation of the physical mixture, with curcumin making up only 12% of the components of the physical mixture, while the rest of the components were the excipients.

### 4.3. Differential Scanning Calorimetry (DSC)

In this study, DSC was used to measure the thermal properties and changes in the physical nature of Formulations A and B. The DSC thermograms of Formulation A (Figure 5d) and its physical mixture (Figure 5e) have shown similar patterns, i.e., the presence of an endothermic peak for Vitamin E TPGs at a range of 33–35 °C, the absence of the endothermic peak of curcumin and the glass transition point of Soluplus^®^. It is speculated that the curcumin and Soluplus^®^ might have dissolved in the molten Vitamin E TPGs during the DSC measurement since Vitamin E TPGs are amphiphilic and only the endothermic peak corresponding to the melting of Vitamin E TPGs was observed in the DSC thermograms of Formulation A and in the physical mixture [62]. The human body temperature is 37 °C, which is above the measured melting point of Vitamin E TPGs. Vitamin E TPGs are very hydrophilic and dissolve quickly in aqueous solutions [63] so, if some curcumin has been dissolved in molten Vitamin E TPGs, they can quickly be released into the intestinal lumen fluid and absorbed by the intestine epithelium as they are already in solution form.

In the thermogram of Formulation B (Figure 6f) and the physical mixture (Figure 6g), only a weak endothermic peak at a range of 44–46 °C was observed, corresponding to the melting point of Poloxamer 407. The endothermic peaks of curcumin and Aerosil®, as well as the glass transition point of MCC, were all absent from the DSC diagram of Formulation B and the physical mixture. A possible explanation is that the absence of the endothermic peak of curcumin might be due to the dissolution of curcumin in the molten poloxamer during the heating ramp; it has already been reported that curcumin is soluble in poloxamers [64]. Aerosil® accounts for less than 1.3% of Formulation B so it is likely that the amount is too little to be detected by DSC. MCC makes up about 70% of Formulation B, so it is unlikely that all of it is dissolved in the molten Poloxamer 407. The disappearance of the glass transition-point peak of MCC in the DSC thermogram of Formulation B might be due to the conversion of MCC from its amorphous state into a crystalline state during the preparation process.

### 4.4. X-ray Powder Diffraction (XRD)

XRD was used to determine the crystallographic structure of Formulations A and B and to detect if the amorphous phase was present in the formulations. When the material is in a crystalline state, it will have a periodic arrangement of its atoms. As a result, the X-rays will only be scattered in certain directions, which leads to highly intense sharp peaks. If a material is in an amorphous state wherein the atoms are randomly arranged, X-rays will be scattered in many directions, leading to a large bump distributed in a wide range, resulting in smooth and broad peaks [65].

The sharp and intense diffraction peaks observed from the XRD spectrum of commercial curcumin (Figure 7a) indicate the significant crystallinity of curcumin. A similar XRD spectrum of curcumin was reported in previous studies [66]. There were no distinct peaks of curcumin and Soluplus^®^, as were observed in the spectrum of Formulation A (Figure 7d). Two distinct sharp peaks were observed at 19.2° and 23.28°, the peak positions and shapes of which were almost identical to the characteristic peaks found in the spectrum of Vitamin E TPGs (peak positions at 19.26° and 23.31°). This showed that the two intense peaks observed in Formulation A were derived from the highly crystalline Vitamin E VTPGs. The absence of all the characteristic peaks for curcumin in Formulation A would suggest that curcumin was very amorphous in the formulation and could be one of the main reasons for the significantly increased solubility of curcumin in Formulation A. As other studies have shown, the solubility of amorphous curcumin was much higher than in their crystalline counterparts, due to the lack of the long-order molecular arrangement of amorphous compounds, which gives them greater molecular mobility and free energy [67]. In contrast, the spectra of the physical mixture of Formulation A showed several sharp peaks that indicate the presence of crystalline curcumin (7.94°, 12.26°, 14.65°, 15.9°, 18.29°, 24.65°, 27.41° and 29.1°), crystalline Soluplus^®^ (8.95°, 17.44° and 25.68°) and crystalline Vitamin E VTPGs (19.25° and 23.31°).

As shown in Figure 8, the XRD spectra of Formulation B and its physical mixture were almost identical to that of MCC. Almost all the characteristic peaks observed in Formulation B and the physical mixture can be found in MCC. No peaks for curcumin, Poloxamer 407, Aerosil® or magnesium stearate were observed. Since MCC makes up approximately 70% of the compositions in Formulation B and the physical mixture, it is very likely that the large amount of MCC masked the curcumin and other excipients, rendering their characteristic peaks undetectable by XRD. As a result, it is inconclusive to say if the curcumin in Formulation B was in a crystalline state or an amorphous state.

### 4.5. Determining Curcumin Content in Commercial Curcumin Powder, Formulation A, Formulation B, Longvida^®^ and Nacumin^®^

The curcumin powders available on the market are a mixture of curcumin, demethoxycurcumin and bisdemethoxycurcumin. Therefore, the actual content of curcumin in the commercial curcumin powder was measured by HPLC before the cytotoxicity assays, in vitro drug transport, and cellular uptake experiments. Information on the curcumin content of Longvida^®^ and Nacumin^®^ was not available from the manufacturers. Therefore, the amount of curcumin in each Longvida^®^ and Nacumin^®^ capsule was measured in this study, along with the curcumin content in Formulation A and Formulation B. From Table 2, it can be observed that approximately 81% of the curcuminoids are from curcumin, which is consistent with the literature value [1]. Among the four test samples, Longvida^®^ had the highest percentage of curcumin at 30%, with no significant difference in the percentage of curcumin regarding the remaining three test samples. Using this information, the amount of each test sample that would be equivalent to 0.1 mM of curcumin concentration (equivalent to 36.84 μg/mL of curcumin) was calculated and is shown in Table 3.

### 4.6. Determination of Cytotoxicity (MTT and LDH Assays)

The MTT assay determines the cell viability in response to extracellular stimuli or therapeutically active agents [68]. In this study, cell viability was used as an indicator for predicting the cytotoxicity of each formulation toward Caco-2 cells. The MTT assay results showed that the cell viabilities were higher than 80% when the curcumin concentration was at 0.08 mM and 0.1 mM, which indicates that these samples, when tested, should not affect the cellular integrity of the Caco-2 cell monolayers [69]. This finding is consistent with a previously reported study [70].

LDH is an enzyme that is normally found within the cell cytoplasm. This enzyme leaks into the cell culture medium when the cells are dead, due to the loss of membrane integrity. In this study, the percentage release of LDH into the cell culture medium was used as the indicator for cell death. Damaged or dead cells that have lost membrane integrity allow the passage of non-permeable molecules and are not suitable for use in the in vitro drug permeation and cell uptake studies. It is, therefore, important to ensure that the samples being tested have a minimal negative impact on the cultured cell line. According to the LDH assay results, the toxicity of all the tested samples to Caco-2 cells was at a minimum when the curcumin concentration was ≤ 0.1 mM. An LDH release of 13.59% was observed for the negative control, which indicated the apoptosis of Caco-2 cells over time [71].

Overall, based on the results of the MTT and LDH assays, 0.1 mM (equivalent to 36.84 μg/mL) of curcumin was chosen as the drug concentration for the preparation of the donor suspension for the cell permeability and drug uptake studies, as this was the highest drug concentration that could be used without damaging the Caco-2 cell monolayers.

### 4.7. Measurement of Transepithelial Electrical Resistance (TEER)

TEER is a universally accepted quantitative technique for measuring the integrity of the tight junction dynamics of the endothelial and epithelial cell monolayers [72]. It was reported that there is a good correlation between the integrity of the Caco-2 cell monolayers and the TEER value [73]. At the end of the drug transport experiment, TEER readings significantly decreased (*p* ≥ 0.05) for all the test samples. However, the decrease in TEER values did not indicate a loss or disruption of cell monolayer integrity, as the difference in TEER values before and after the experiment was less than 100.Ω cm^2^, which was within acceptable limits [74]. In addition, the average TEER readings for all samples tested were above 200 Ω cm^2^, meaning that the Caco-2 cell monolayer remained intact, limiting the diffusion of non-permeable material across the barrier [72]. In previously published studies, it was observed that the TEER readings decreased after the in vitro cell permeability experiments, while no significant cytotoxicity of the test compounds was seen via MTT and LDH testing [36,74,75].

### 4.8. In Vitro Drug Transport Study

In this study, bidirectional Caco-2 cell in vitro drug permeation experiments were conducted to mimic the intestinal absorption of curcumin and determine the intestinal permeability of curcumin. In the A-B drug permeation experiments, the donor suspensions of the tested compounds were placed on the apical side of a Caco-2 cell monolayer and the amount of curcumin transported to the basolateral side of the cells was measured to assess the drug permeability of the monolayer for each tested sample. In the B-A drug permeation experiments, the tested samples were added to the basolateral side and the amount of the drug transported to the apical side was measured, to test for the presence of active transport or efflux across the Caco-2 cell monolayer.

The permeation experiments were stopped after 3 h because the average small intestine transit time for a pharmaceutical dosage form has been reported to be 3 h [76]. The experiments were designed to mimic the absorption of the drug in the human small intestine. As curcumin is unlikely to remain in the small intestine for longer than 3 h, there was no need to perform experiments beyond this time period.

From this comparison, it was found that the concentrations of curcumin permeation from A to B were significantly higher than those from B to A for all the tested samples (excluding commercial curcumin) (*p* ≤ 0.05). This suggests that the drug influx might be more predominant than the drug efflux across the intestinal epithelium for all the tested samples (except for commercial curcumin). To find more evidence to support this view, the efflux ratio of each test sample was calculated. Formulation A had an efflux ratio of 0.38, while other tested formulations have shown an efflux ratio of 0, which indicated that drug efflux was not involved in the drug transport; this could possibly be due to an inhibition of the apical efflux transporters that might have restricted B-A drug transport but could increase the intracellular drug accumulation [77]. For an efflux ratio value of greater than 0 but less than 1, indicates that the absorptive influx transport is more dominant than efflux transport during the drug transport process across the Caco-2 cell monolayer [78].

It has been reported that curcumin is absorbed in the small intestine, predominantly by means of passive diffusion [79]. Thus, the intestinal permeation of curcumin can generally follow Fick’s first law, wherein the rate of diffusion of a solute in solution across the unit area (normally a surface or membrane) is proportional to the concentration gradient [80]. According to this theory, as the concentration of dissolved curcumin increases in the intestine lumen fluid, more curcumin molecules can be present at the surface of the intestinal epithelial cells and more drugs per unit time per surface area (flux) can permeate across the cells [81]. In this study, it was found that the tested sample with a higher value of C_0_ (suspension) (A-B) also showed a higher concentration of A-B curcumin permeation and Papp (Solution) (A-B) values. Since the values of C_0_ (suspension), (A-B) is associated with the aqueous solubility of curcumin; this suggests that there was a positive correlation between the aqueous solubility and the permeability of curcumin across the Caco-2 cell monolayer. The positive correlation between the drug aqueous solubility and drug permeability was also found in other studies [28,82,83].

Formulation A, which contains Soluplus^®^ and Vitamin E TPGs, converted the crystalline curcumin into an amorphous form, as shown by the XRD results. It was reported that the conversion of crystalline drugs into an amorphous form can contribute to increasing the drug solubility [84,85,86]. In a similar study, a solid formulation made from Soluplus^®^/Vitamin E TPGs was reported to show the conversion of insoluble valsartan from a crystalline to an amorphous state, which increased the aqueous solubility of the drug and ultimately led to a significant increase in oral bioavailability [12]. Formulation A showed the highest curcumin A-B permeation concentration (0.49 ± 0.07 μg/mL, equivalent to 0.00133 ± 0.00019 mM), Papp (solution), (A-B) value (1.12 ± 0.15 × 10^−6^ cm/s) and C_0_ (suspension), (A-B) value (17.00 ± 2.95 μg/mL). It is clear that the permeability of curcumin from A to B benefits from the increased solubility. The ability of Formulation A to improve the solubility of curcumin has been demonstrated by the solubility study results. Apart from the aqueous solubility, inhibition of the efflux transporters of curcumin can also help to improve permeability. Vitamin E TPGs, one of the excipients of Formulation A, was reported to have the ability to inhibit the apical efflux pumps of curcumin, such as P-gp [62], which produce less intracellular curcumin to be pumped back to the apical side of the Caco-2 monolayer.

Formulation B showed the second-highest A-B curcumin permeation concentration (0.11 ± 0.03 μg/mL) and Papp (solution) (A-B) value (0.24 ± 0.07 × 10^−6^ cm/s). It also showed the second-highest value of C_0_ (suspension) (A-B) (3.72 ± 0.71 μg/mL). Interestingly, Nacumin^®^ showed similar A-B curcumin permeation concentrations (0.09 ± 0.01 μg/mL) to Formulation B. However, it has a much lower C_0_ (suspension) (A-B) value (0.40 ± 0.05 μg/mL) than that of Formulation B. This suggests that the improved drug permeability shown by Nacumin^®^ was more likely to be related to factors other than solubility, such as particle size reduction. Longvida^®^ exhibited the lowest A-B curcumin permeation concentration (0.04 ± 0.00 μg/mL, equivalent to 0.00011± 0.00001 mM), Papp (solution), (A-B) value (0.09 ± 0.01 × 10^−6^ cm/s) and C_0_ (suspension), (A-B) (0.31 ± 0.03 μg/mL) among all the tested samples (excluding commercial curcumin). This further confirmed the positive correlation between the solubility and the permeability of curcumin. No drug permeation was found from A to B and B to A for commercial curcumin. As shown in Table 5, it exhibited extremely low drug solubility in the donor chamber, which was believed to be the main reason for zero drug permeation.

### 4.9. Cellular Uptake Test

The results of the cellular drug uptake test provided a prediction for the amount of drug that has been absorbed by the intestinal epithelial cell but has not yet been released to the blood circulation (basolateral side) or effluxes back to the intestinal lumen (the apical side) [87,88].

In this study, it was found that the cellular uptake of curcumin absorbed from the apical side was associated with C_0_ (suspension) (A-B). Formulation A showed both significantly higher C_0_ (suspension) (A-B) (17.00 ± 2.95 μg/mL) and apical cellular uptake of curcumin (0.46 ± 0.09 μg/mL) than all other tested samples (*p* ≤ 0.05). Next, Formulation B exhibited the second-highest C_0_ (suspension), (A-B) (3.72 ± 0.71 μg/mL) and apical cellular uptake of curcumin (0.10 ± 0.02 μg/mL); the values were significantly higher than Nacumin, Longvida^®^ and commercial curcumin (*p* ≤ 0.05). In contrast, Nacumin^®^, Longvida^®^ and commercial curcumin all exhibited the lowest apical cellular uptake of curcumin and C_0_ (suspension) (A-B). There were no significant differences between the three tested samples, in terms of the apical cellular uptake of curcumin (*p* > 0.05), as well as the C_0_ (suspension) (A-B) (*p* > 0.05). Since C_0_ (suspension) (A-B) is correlated to the aqueous solubility of curcumin, this suggests that the apical cellular uptake of curcumin can be affected by solubility. This view was further supported by other published studies, which found a positive correlation between the concentration of the drug dissolved in the donor compartment and the in vitro apical cellular drug uptake of Caco-2 cells [89,90].

It was unclear whether the uptake of curcumin by the cells from the basolateral side was influenced by solubility. Formulation A showed the highest C_0_ (suspension) (B-A) values (15.51 ± 1.46 μg/mL) among all the tested samples and also exhibited the highest basolateral cellular uptake of curcumin (0.12 ± 0.04 μg/mL). Conversely, Formulation B has shown significantly higher C_0_ (suspension) (B-A) (1.88 ± 0.13 μg/mL) compared to Nacumin^®^ and Longvida^®^, but they all showed a very similar basolateral cellular uptake of curcumin, with no significant differences (*p* > 0.05).

It is worth noting that Nacumin^®^ exhibited higher curcumin uptake from the basolateral side than the apical side, but no curcumin permeation from B-A was detected. This might suggest that there was an inhibition of apical efflux transporters that prevented intracellular curcumin from being pumped out on the apical side of the cell monolayer.

## 5. Conclusions

In this study, in vitro drug permeation experiments and cellular uptake tests (using Caco-2 cell monolayers as the model) were carried out to simulate drug absorption across the intestinal epithelial cells. The results of the cytotoxicity assays (MTT and LDH assays) and TEER measurement have shown that all the tested samples were well tolerated by the Caco-2 cell monolayers at the concentrations tested (0.1 mM, equivalent to 36.84 μg/mL). Among all the tested samples, Formulation A has shown the highest permeability of curcumin across the Caco-2 cell monolayer, as well as of the cellular uptake of curcumin, which was attributed to the increased solubility of curcumin and the potential inhibition of apical efflux transporters by Vitamin E TPGs. In conclusion, Formulation A is a promising solid formulation that has the potential to increase the oral bioavailability of poorly water-soluble compounds, such as curcumin.

## Figures and Tables

**Figure 1 biomolecules-12-01739-f001:**
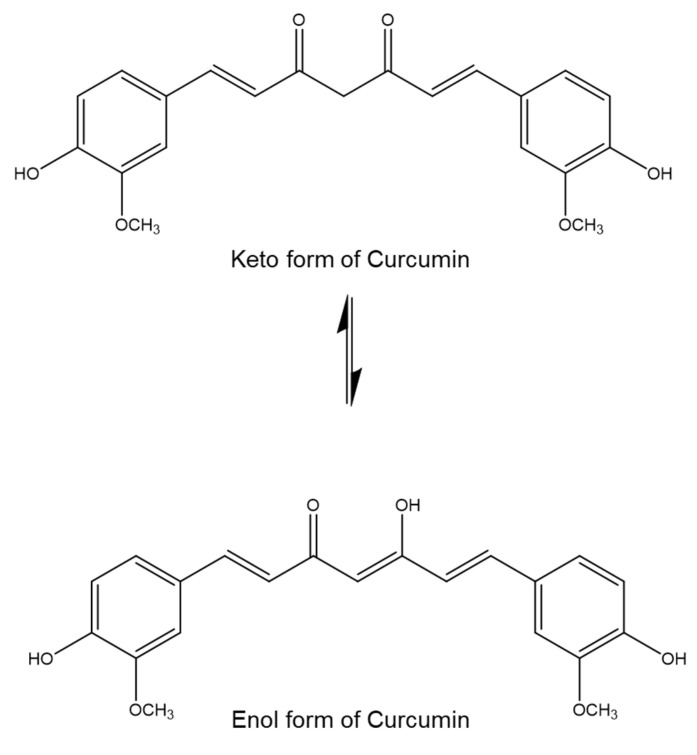
Chemical structure of curcumin.

**Figure 2 biomolecules-12-01739-f002:**
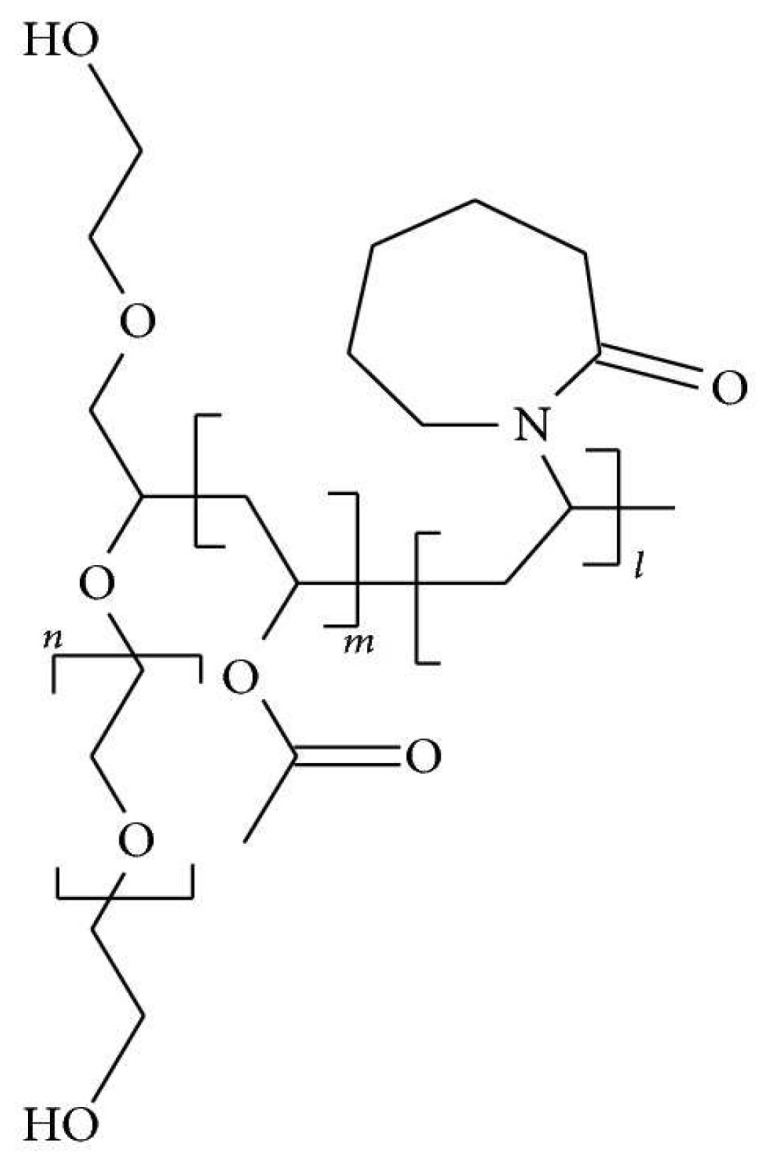
Chemical structure of Soluplus^®^.

**Figure 3 biomolecules-12-01739-f003:**
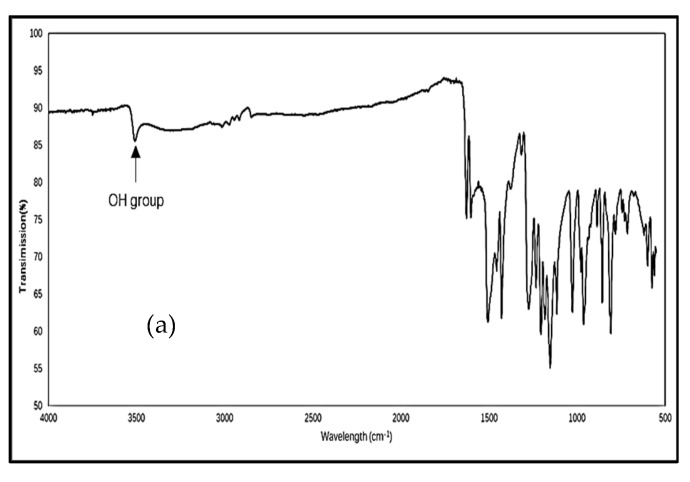
FTIR spectra of (**a**) commercial curcumin, (**b**) Soluplus^®^, (**c**) Vitamin E TPGs, (**d**) Formulation A and (**e**) physical mixture of Formulation A (curcumin:Soluplus:Vitamin E TPGs 1:10:10).

**Figure 4 biomolecules-12-01739-f004:**
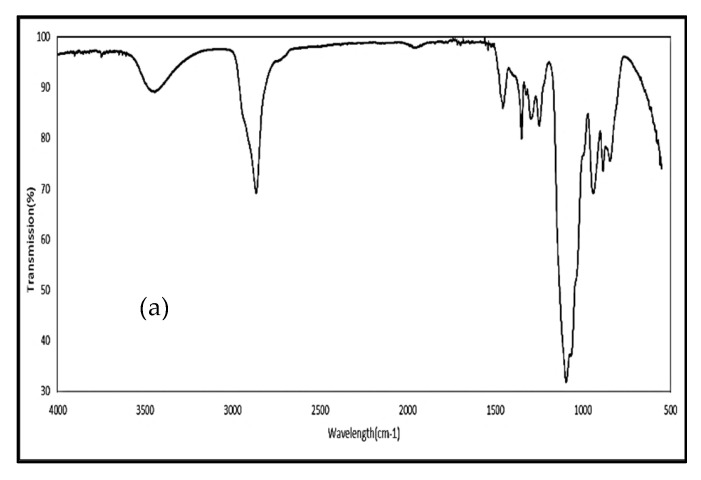
FTIR spectra of the (**a**) PEG400, (**b**) Poloxamer 407, (**c**) MCC, (**d**) Aerosil®, (**e**) magnesium stearate, (**f**) Formulation B, and (**g**) the physical mixture of Formulation B (curcumin: PEG400: Poloxamer 407: MCC: Aerosil®: magnesium stearate 1:0.5:0.9:5.6:0.1:0.1).

**Figure 5 biomolecules-12-01739-f005:**
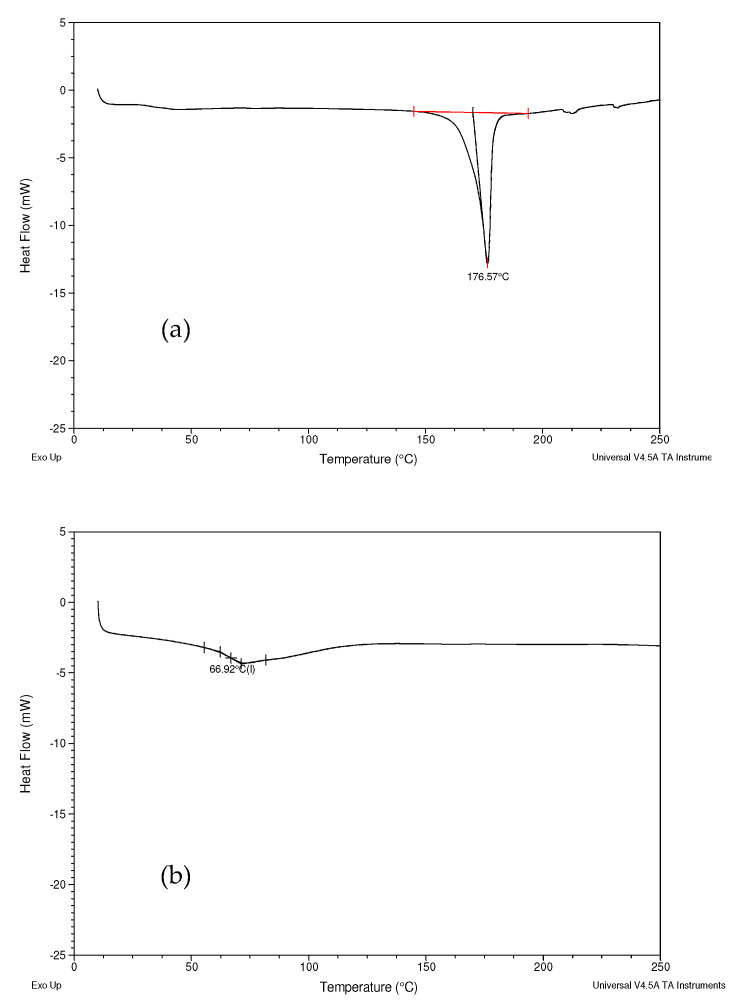
DSC thermograms of (**a**) commercial curcumin, (**b**) Soluplus^®^, (**c**) Vitamin E TPGs, (**d**) Formulation A (**e**) the physical mixture of Formulation A (curcumin: Soluplus: Vitamin E TPGs 1:10:10), showing the endothermic peaks.

**Figure 6 biomolecules-12-01739-f006:**
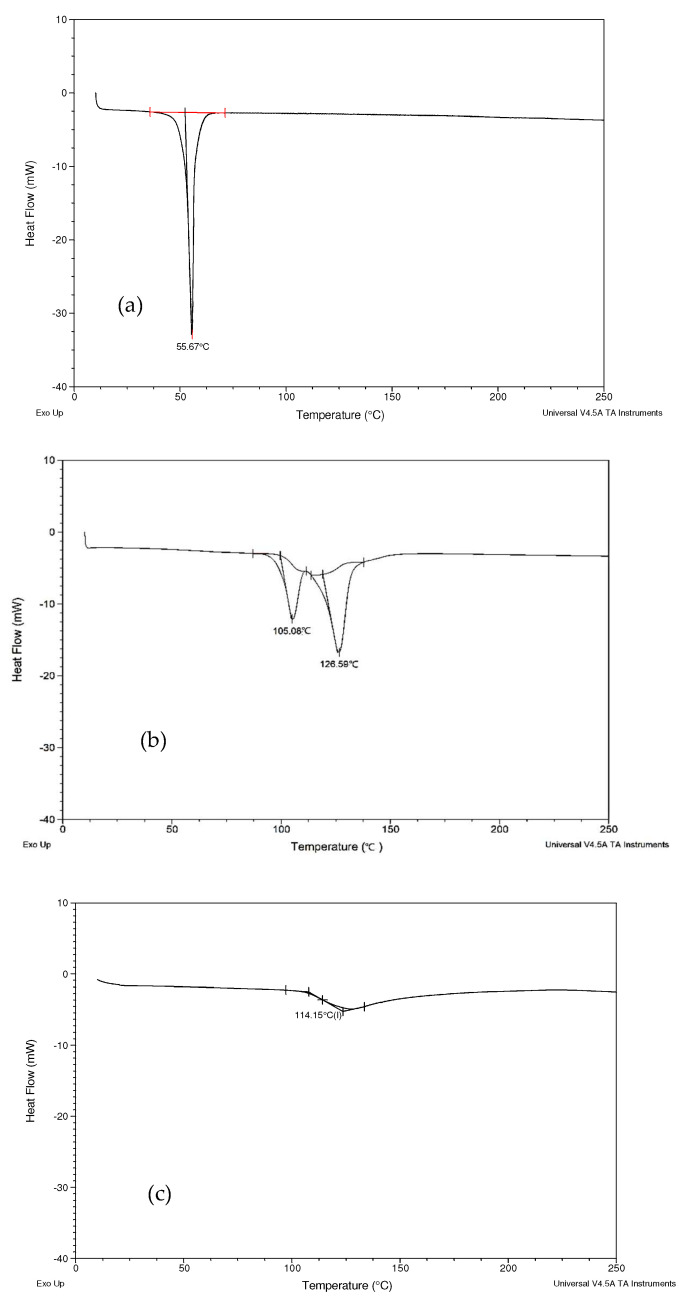
DSC thermograms of (**a**) Poloxamer 407, (**b**) magnesium stearate, (**c**) MCC, (**d**) PEG400, (**e**) Aerosil®, (**f**) Formulation B, (**g**) a physical mixture of Formulation B (commercial curcumin: PEG 400: Poloxamer 407: MCC: Aerosil®: magnesium stearate 1:0.5:0.9:5.6:0.1:0.1), showing the endothermic peaks.

**Figure 7 biomolecules-12-01739-f007:**
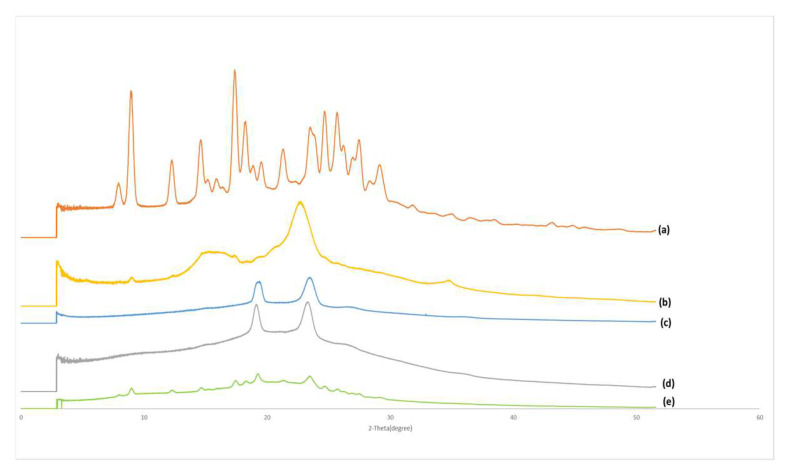
XRD spectra of (**a**) commercial curcumin powder, (**b**) Soluplus^®^, (**c**) Vitamin E TPGs, (**d**) Formulation A, and (**e**) the physical mixture of Formulation A.

**Figure 8 biomolecules-12-01739-f008:**
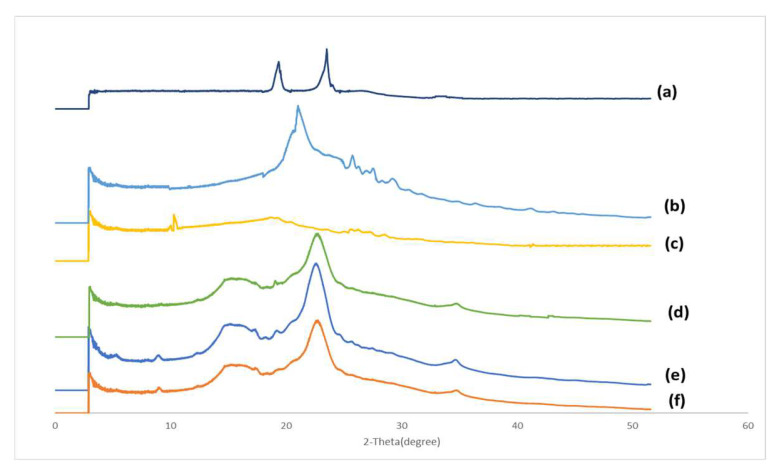
XRD spectra of (**a**) poloxamer 407, (**b**) Aerosil®, (**c**) magnesium stearate, (**d**) MCC, (**e**) Formulation B, and (**f**) the physical mixture of Formulation B.

**Figure 9 biomolecules-12-01739-f009:**
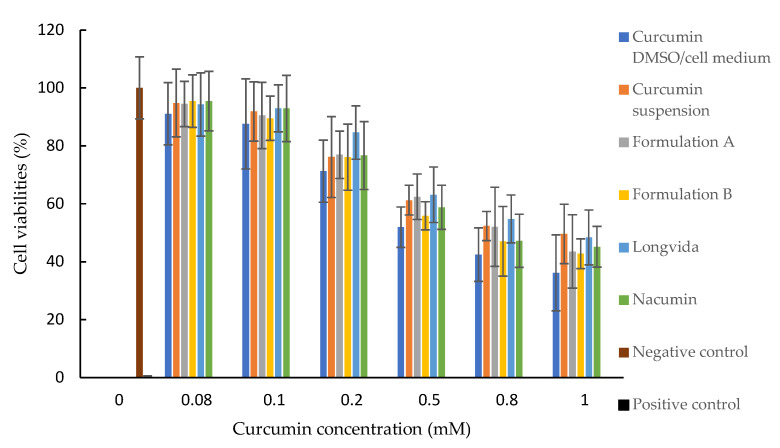
Cell viability (MTT assay) of Caco-2 cells when exposed to each formulation, with equivalent curcumin concentrations from 0.08 to 1 mM (*n* = 18, mean ± SD).

**Figure 10 biomolecules-12-01739-f010:**
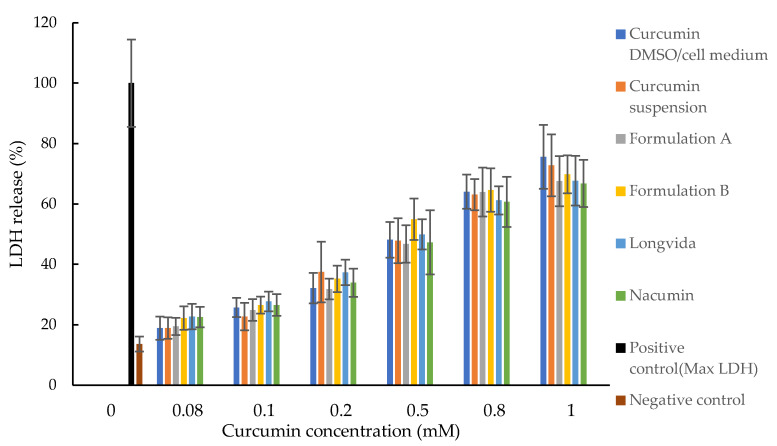
The percentage of LDH released from Caco-2 cells when exposed to each formulation, with equivalent curcumin concentrations from 0.08 to 1 mM (*n* = 18, mean ± SD).

**Figure 11 biomolecules-12-01739-f011:**
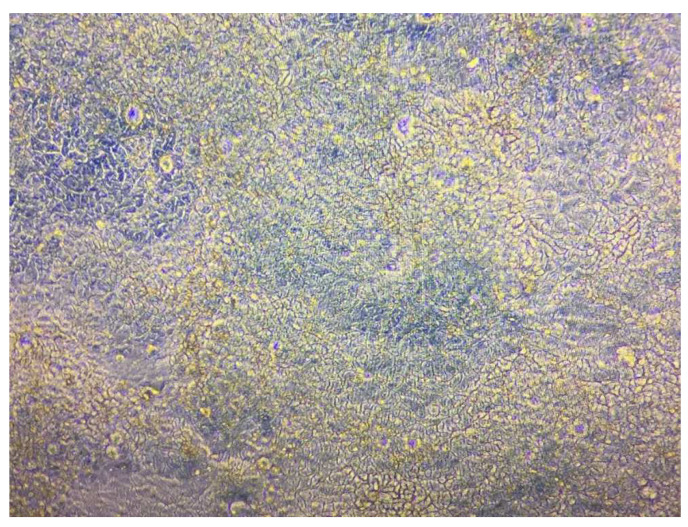
Light microscope image of a well-formed Caco-2 cell monolayer after 21 days of cell culture (20× magnification).

**Figure 12 biomolecules-12-01739-f012:**
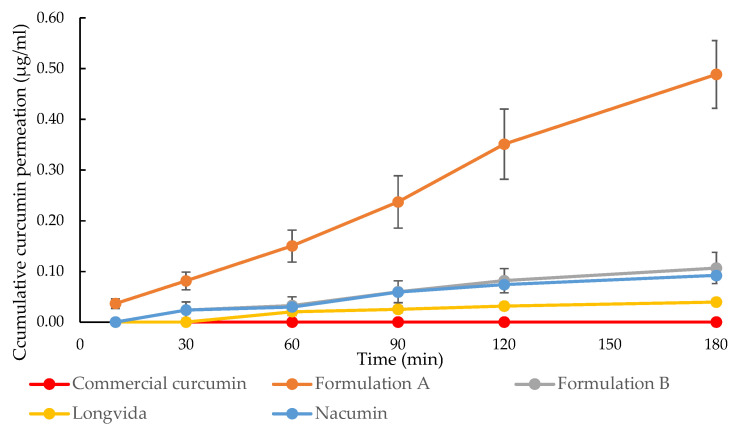
Cumulative amounts of curcumin passing through the Caco-2 cell monolayers, from the apical to the basolateral chambers, A-B (*n* = 9, mean ± SD).

**Figure 13 biomolecules-12-01739-f013:**
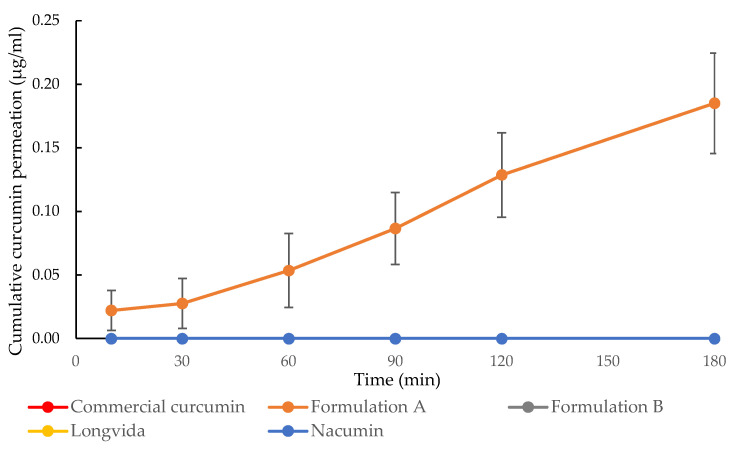
Cumulative amounts of curcumin passing through the Caco-2 cell monolayers from the basolateral to the apical chambers, B-A (*n* = 9, mean ± SD).

**Figure 14 biomolecules-12-01739-f014:**
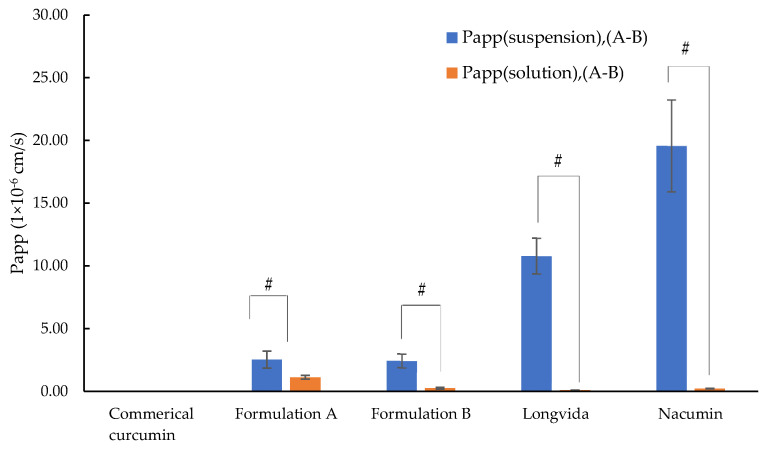
Comparison of Papp (suspension), (A-B) and Papp (solution), (A-B) (*n* = 9, mean ± SD). # Denotes statistically significant differences between Papp (suspension), (A-B) and Papp (solution), (A-B) of each formulation (*p* ≤ 0.05).

**Figure 15 biomolecules-12-01739-f015:**
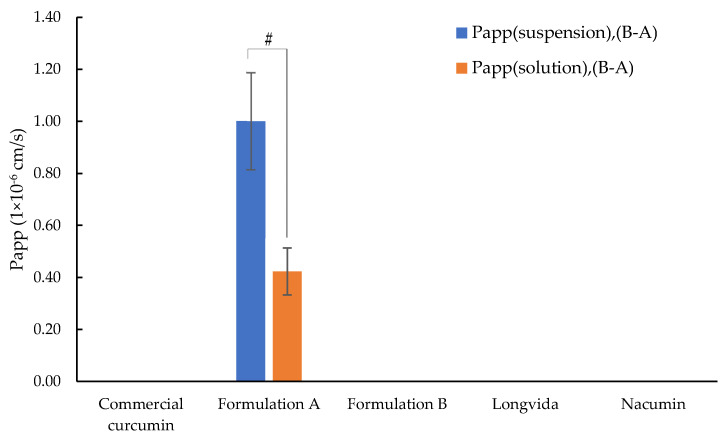
Comparison of Papp (suspension), (B-A) and Papp (solution), (B-A) (*n* = 9, mean ± SD) # Denotes statistically significant differences between Papp (suspension), (B-A) and Papp (solution), (B-A) of each formulation (*p* ≤ 0.05).

**Figure 16 biomolecules-12-01739-f016:**
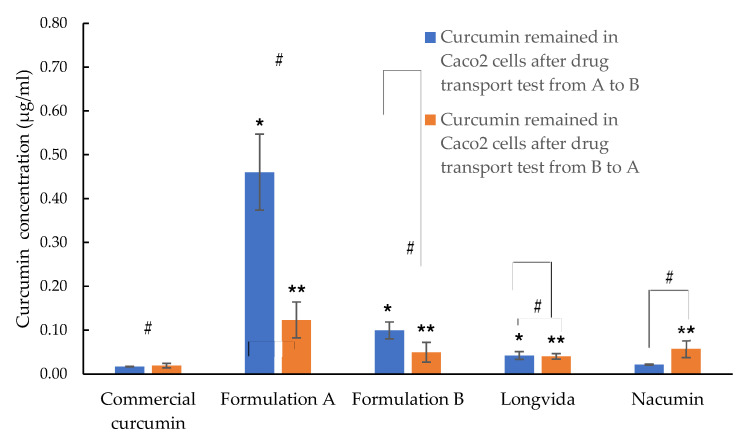
Curcumin accumulation in the Caco-2 cell after the drug transport experiment from the A to B and B to A directions (*n* = 9, mean ± SD). ***** Denotes the statistically significant difference in curcumin cellular accumulation between a formulation sample and commercial curcumin from A to B. ****** denotes the statistically significant difference in curcumin cellular accumulation between a formulation sample and commercial curcumin, from B to A. Note: **#** denotes a statistically significant difference between the curcumin cellular accumulation of a test sample from A to B and B to A (*p* ≤ 0.05).

**Table 1 biomolecules-12-01739-t001:** The amount of each ingredient required for preparing 1 mL of the LDH substrate mixture.

Ingredient	Amount/mL
L-lactate	2.5 mg
ß-NAD	2.5 mg
MTT	0.25 mg
MPMS	0.034 mg
Tris-HCL	0.9 ml
1% (*v*/*v*) Triton-X100	0.1 ml

**Table 2 biomolecules-12-01739-t002:** Saturation solubility of curcumin from Formulation A, Formulation B and commercial curcumin in pH 6.8 buffer (*n* = 3, mean ± SD).

	Curcumin Solubility (μg/mL)
Formulation A	70.33 ± 10.26
Formulation B	25.00 ± 0.60
Commercial curcumin	0.08 ± 0.01

**Table 3 biomolecules-12-01739-t003:** The percentage of curcumin contained in commercial curcumin powder, Formulation A, Formulation B, Longvida^®^ and Nacumin^®^ (mean ± SD, *n* = 6).

	Curcumin (% *w*/*w*)
Commercial curcumin powder	81.46 ± 2.19
Formulation A	6.74 ± 0.45
Formulation B	8.91 ± 0.85
Longvida^®^	30.67 ± 1.33
Nacumin^®^	7.15 ± 0.15

**Table 4 biomolecules-12-01739-t004:** The concentration of each test sample needed to prepare a donor suspension containing 0.1 mM curcumin (0.1 mM, equivalent to 36.84 μg/mL of curcumin).

	Sample Concentration (μg/mL)
Commercial curcumin powder	45.22
Formulation A	545.59
Formulation B	413.46
Longvida^®^	120.12
Nacumin^®^	515.24

**Table 5 biomolecules-12-01739-t005:** Average TEER readings of the Caco-2 cell monolayers at the start and end of the in vitro drug transport experiment (Ω cm^2^) (*n* = 9; mean ± SD).

	At t = 0 min	After t = 180 min
Curcumin	328 ± 23	271 ± 23
Formulation A	330 ± 28	270 ± 28
Formulation B	316 ± 56	270 ± 50
Longvida	313 ± 48	257 ± 35
Nacumin	330 ± 57	271 ± 58

**Table 6 biomolecules-12-01739-t006:** Initial curcumin concentration dissolved in donor chamber (C_0_ suspension) (*n* = 9, mean ± SD).

	C_0_ (Suspension), (A-B) (μg/mL)	C_0_ (Suspension), (B-A) (μg/mL)
Curcumin	0.07 ± 0.01	0.04 ± 0.01
Formulation A	17.00 ± 2.95	15.51 ± 1.46
Formulation B	3.72 ± 0.71	1.88 ± 0.13
Longvida	0.31 ± 0.03	0.23 ± 0.04
Nacumin	0.40 ± 0.05	0.31 ± 0.03

**Table 7 biomolecules-12-01739-t007:** Ranking of samples in terms of Papp (A-B) value (from high to low).

	Papp (Suspension), (A-B) (cm/s)	Papp ( Solution), (A-B) (cm/s)
1st	Nacumin^®^ (19.58 ± 3.66 × 10^−6^)	Formulation A (1.12 ± 0.15 × 10^−6^)
2nd	Longvida^®^ (10.79 ± 1.42 × 10^−6^)	Formulation B (0.24 ± 0.07 × 10^−6^)
3rd	Formulation A (2.52 ± 0.68 × 10^−6^)	Nacumin^®^ (0.21 ± 0.03 × 10^−6^)
4th	Formulation B (2.43 ± 0.55 × 10^−6^)	Longvida^®^ (0.09 ± 0.01 × 10^−6^)
5th	Commercial curcumin (0)	Commercial Curcumin (0)

## Data Availability

Not applicable.

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
