# Peer review of "Increasing Cellular Uptake and Permeation of Curcumin Using a Novel Polymer-Surfactant Formulation"

_biomolecules, 2022, doi:10.3390/biom12121739_

Round 1

Reviewer 1 Report

The manuscript "Increasing cellular uptake and permeation of curcumin using a novel polymer-surfactant formulation" presented to authors Liu Z et al, describes the potential of novel polymer- and surfactant-based formulations (A and B) to be able to enhance and improve the cellular permeability of poorly water-soluble bioactive molecules. The bioactive molecule chosen and studied by the authors was curcumin.  The experimental model used by the authors to prove their thesis was a monolayer model of Caco-2 cells in vitro. The results of cytotoxicity assays (MTT and LDH as it says) and TEER measurement showed that all tested samples were well tolerated by monolayers of Caco-2 cells at the tested concentrations. The best of all was formulation A. The authors consider this, therefore, a promising solid formulation useful in the pharmaceutical field for making and developing formulations for the oral administration of poorly water-soluble compounds such as curcumin.

This manuscript deals with a very interesting topic and one that opens up considerable scope for future research and exploration by the reader.

The manuscript is clearly, correctly, and accurately produced in most of its sections.

The authors have chosen to present data, both in quantity and specificity, well organized to prove their thesis.

However, I believe that some minor revisions need to be made by the authors for the manuscript to be published.

Minor Revision

Equations: For graphic clarity, I urge the authors to improve all equations presented, unifying them in font and size.

Table 1: I urge the authors, for greater clarity and better use by the reader, to present the data on the composition of the LDH substrates mixture solution by expressing all concentrations as "Amount /ml" and arranging them in the table in descending order of quantity.

Figures:  All figures submitted, except figure 3, must all be resized (they are too large), and they must all be revised graphically. They all need to be unified in style (get rid of the inner frames), change the font (make it uniform with the text of the manuscript), change the color of the text and axes ( use black and not gray), insert all major ticks intervals on the y-axes. The legends in Figures 6-7-8 should be inserted within the graph.

Kind Regards

Reviewer 2 Report

The authors described a novel approach to prepare a polymer-surfactant formulation of curcumin. In this research, curcumin powder was prepared using two different methods. The authors displayed good data and rigorous in vitro experimentation to support and demonstrate the potential of this system to improve the permeability of an in vitro Caco-2 cell monolayer model to curcumin. However, there are several comments that need to be addressed prior to publication:

1. Authors need to evaluate the properties of the powder from each formulation. Some data from XRD, FTIR, DSC can be useful to support the discussion and conclusion of this manuscript.

2. Saturation solubility of curcumin from formulation A and B need to be determined in order to show an improvement in curcumin solubility.

Reviewer 3 Report

The manuscript entitled "Increasing cellular uptake and permeation of curcumin using a 2 novel polymer-surfactant formulation", from the authors Zhenqi Liu, Alison B Lansley, Tu Ngoc Duong, John D Smart and Ananth S Pannala.

In general, the manuscript is excellent. The experiments are well planned, the methods and tests are well chosen and the analysis is well done.

1. In the part of the manuscript "2.2. Preparation of Formulations A and B" it is written: "... and acetone was removed under pressure (40 °C, 250 mbar, 200 125 rpm)." - (Lines 125 and 126). Obviously, the acetone was removed under vacuum (250 mbar). Please correct this sentence.

2. In the part of the manuscript "2.3. Determining curcumin content in commercial curcumin powder, Formulation A, 145 Formulation B, Longvida® and Nacumin®" it is written: "... (SphereClone™, 150 x 46 mm, 5 μm particle size). " - (Line 159). Probably referring to the 150 x 4.6 mm, 5 μm particle size column. Please correct this sentence.

I consider that manuscript should be published in journal „Biomolecules“ after minor corrections.

Reviewer 4 Report

The authors Z. Liu and co-workers submitted the manuscript entitled „Increasing cellular uptake and permeation of curcumin using a novel polymer-surfactant formulation” to the journal “Biomolecules” in order to be considered for publication as an “Article”.

The study reports on the permeation and uptake (into Caco2 cells) of different species/formulations of curcumin. In particular, the authors investigated two commercially available products and two formulations A and B. The experimental procedure also covered to study the cytotoxicity (MTT and LDH assay) as well as TEER measurements. Finally, the authors conclude that Formulation A gives promise for an increased bioavailability of curcumin. These results could be extended to further drugs with poor water solubility.

The authors provide a comprehensive study. However, there are some aspects the authors should address to strengthen their study.

It is suggested to add the chemical structures of curcumin and Soluplus® to the introduction.

For an excipient to be useful with an active ingredient, it must be ensured that they do not react with each other. Can the authors include a short sentence with reference in their introduction which excludes the interaction of the two substances curcumin and Soluplus.

Maybe the authors can make a short mentioning in their manuscript the field of application of Longvida and Nacumin. Why is supplementing with curcumin valuable?

“Calibration plots showed a linear relationship with a correlation coefficient of R2 > 0.9995 over the entire range.” Can the authors please provide this in a kind of supplementary material?

PBS: phosphate-buffered saline?

TEER was performed after 21 d of incubation. Can the authors provide more detailed information on the cell culture, e.g. change of medium in the meantime.

Was a wash step performed in the cell uptake studies? After centrifugation and before extraction with DMSO? Curcumin on the cell surface, which has not yet been taken up into the Caco2 cells, could be removed in this way. However, this does not seem to be the case with the described procedure.

Also, in general: What is known about the uptake pathways of curcumin into the cell? Passive diffusion due to lipophilicity or maybe a transporter can be exploited? The authors should include this in their manuscript, e.g., in the introduction.

@ Determining the curcumin content using HPLC: Can the authors provide the chromatograms, e.g. in terms of supplementary material. Is it possible to separate curcumin, demethoxycurcumin, and bisdemethoxycurcumin using this methods? Where are the methods from (literature), or have they been developed in the course of the current project. The authors should comment on that in their manuscript.

@ MTT assay. The authors used a negative control. How about a positive control (LDH-assay: better term it positive control, not high control)? This is missing in the shown dataset. Please add.

@ Figure 4: The uptake into cancer cells seems not completed after 3 h (180 min). Studies from the group of Gust published e.g. in Dalton Trans documents that uptake is finished after 4-6 h. Why the authors did stopped their experiments as soon as after 3 h of incubation? Please provide a reason in the manuscript.

@ Figure 5: It is confusing to use the yellow color for four different species. Please revise.

@ Table 6: Maybe the authors use some kind of scale to order the Papp, considering the real values?

Although there will be a close editing by the MDPI publisher, the authors are kindly asked to develop on formal errors and inconsistency, such as [12-15] instead [12,13,14,15], case shift (e.g., lines 100-105), use of space, 0.5 g of …/ 5 g of …, & is not scientific writing, different fonts, introduction and consistent use of abbreviations, use of dash, among others.

In general, the manuscript and the subject matter covered therein correspond to the scope of the special issue “Recent Advances in Natural Products Research as Therapeutic Agents: Focus on Treatment of Cancer”.

All the best!

Round 2

Reviewer 2 Report

The authors have effectively addressed all reviewer concerns and improved this manuscript. Acceptance is recommended.

Reviewer 4 Report

The authors provided a revised version of their manuscript “Increasing cellular uptake and permeation of curcumin using a novel polymer-surfactant formulation”. They acted on every suggestion and concern. Further processing of the manuscript is therefore suggested. All the best!